# A study on the influencing factors of the public's willingness to donate funds for critical illness crowdfunding projects on network platforms

**Lu Chen**[1,2]*, **Fan Luo**[3]*, **Wanshi He**[4]*, **Heng Zhao**[5]*, **Liru Pan**[1]

**1** School of Economics and Management, East China Jiaotong University, Nanchang, China, **2** School of Management, Nanchang University, Nanchang, China, **3** Department of Contemporary Chinese Studies, The Chinese University of Hong Kong, Hong Kong, China, **4** Business School, Shantou University, Shantou, China, **5** Business School, University of Shanghai for Science and Technology, Shanghai, China

* melody1423@foxmail.com (LC); 478479051@qq.com (FL); 3224240976@qq.com (WH); zhaoheng199821@163.com (HZ)

**Data Availability Statement:** Replication materials to recreate all analyses in this manuscript are available in the Open Science Framework database (https://osf.io/wq8z5/). DOI 10.17605/OSF.IO/WQ8Z5 We have uploaded the minimum

## Abstract

As an emerging charity model, critical illness network crowdfunding provides a source of funds for some critically ill patients in China who have difficulty paying their high treatment costs by themselves. This study aims to investigate the influencing factors of the public's willingness to donate to critical illness crowdfunding projects on Internet platforms. From a perspective combining the technology acceptance model and the theory of planned behavior, a complex and comprehensive structural equation model is proposed. We randomly selected 1,000 members of the public in China and empirically verified the study framework through structural equation modeling (SEM) based on 710 valid questionnaires. The results show that the public's donation willingness and the social distance to a critical illness crowdfunding project on an online platform positively affect the public's donation behavior, and donation attitude positively affect donation willingness; perceived usefulness and empathic concern positively affect the public's donation attitude, which in turn affects its donation willingness. This study confirms that members of the public are more likely to help people who have similar demographic factors or people who are similar to themselves and have the same values, i.e., people who have a close social distance. It innovatively proposes and verifies the hypothesis that empathic concern can significantly positively affect users' perceived usefulness and donation attitude. Strong empathic concern triggers donation willingness and behavior.

## Introduction

With the rapid development of the Internet in recent years, the diversified network applications and services of China's "Internet +" have flourished, promoting the transformation of Chinese society and, simultaneously, providing more opportunities to identify people who highly agree with the mission or beliefs of organizations [1].

anonymous data set, which includes the values behind the reported average, standard deviation and other measures, as well as the values used to build the graph(in Tables).

**Funding:** This research project was supported by the Jiangxi Province Department of Education Science and Technology Project (Grant No. GJJ180331).LC and FL conceived and designed the research and methodology. LC provided guidance throughout the entire research process. FL and WH collected and compiled all of the measurements and literature. FL and WH completed the calculation and analyzed the results. LC proposed put forward the policy recommendations. HZ had critically revised the manuscript critically for important intellectual content. HZ supplemented the English manuscript. LP revised and approved the manuscript. We would like to thank AJE (https://www.aje.com/) for English language editing.

**Competing interests:** The authors have declared that no competing interests exist.

As an emerging social media-based online financing approach, network crowdfunding allows project fundraisers to apply for small amounts of money from the public distributed online using social media and new payment technologies to support valuable projects [2]. Network crowdfunding usually involves four types of fundraising: product crowdfunding, bond crowdfunding, equity crowdfunding and charitable crowdfunding (also known as donation crowdfunding). The emergence of charitable crowdfunding means that this new mode of financing, which began with commercial financing, has begun to enter the charity field [3].

Critical illness network crowdfunding is an emerging charity model that organically combines charitable donations and social media. It is characterized by low entry barriers, simple operations, and a high cost-benefit ratio. It provides services for critically ill patients who cannot afford to pay their high treatment costs by themselves. The provision of funds avoids delays in treatment or abandonment of treatment due to insufficient funds. In 2019, in addition to lottery public welfare funds and large donations from philanthropists, Internet donations were the main source of domestic charity resources. The "Charity Blue Book: China Philanthropy Development Report" (2020) shows that in the first half of 2019, 20 Internet public fundraising information platforms designated by the Ministry of Civil Affairs issued more than 17,000 pieces of fundraising information for more than 1,400 public fundraising charitable organizations across the country. With a total of 5.26 billion clicks, the attention and participation of people raised more than 1.8 billion yuan. In addition, the "99 Charity Day" in 2019 reached a new high in fundraising, transparency and coverage. Forty-eight million caring netizens donated 1.783 billion yuan through Tencent's charity platform, and more than 2,500 companies donated 307 million yuan. With a matching donation of 399.99 million yuan provided by the Tencent Charity Foundation, a total of 2.49 billion yuan was raised this year on the "99 Charity Day". While many netizens have shown their support, there are also many risk factors involved in the rapid development of critical illness crowdfunding platforms. According to a survey of 1,737 people conducted by the China Youth Daily Social Survey Center in 2015, 47.4% of respondents had participated in donations through online platforms, but only 28.5% trusted the charitable organizations or other types of organizations in online donations. Regarding the individuals involved in such fundraising, 62.4% of respondents worried about the potential risks of donation fraud and other types of fraud in online fundraising [4].

Recently, some well-known online public welfare critical illness crowdfunding platforms have become embroiled in scandals involving donation fraud and other types of fraud. Such scandals stand in stark relief to the sincere aid texts of individual help-seekers when applying for public welfare crowdfunding, challenging the public to address donation fraud. The psychological expectation of zero tolerance for donation fraud has caused the public to have low recognition of critical illness network crowdfunding, frustrating the achievement of participation in online critical illness crowdfunding [5]. Given the social trend of charity commercialization and public welfare marketing, the public cannot monitor and follow up on the progress of donations after a one-click donation, and the use and destination of donations are unknown. The problems caused by the imperfections in the development of critical illness network crowdfunding warrant academic attention and research.

What factors influence the public's willingness to donate to critical illness crowdfunding projects? How should critical illness crowdfunding platforms optimize this process, expand their functions, perfect the systems involved, and realize benign operations to help more help-seekers? To answer these questions, this study, based on stakeholder theory and social identity theory and from a perspective combining the technology acceptance model (TAM) and the theory of planned behavior(TPB), proposes a model for identifying the influencing factors of the public's willingness to donate to critical illness crowdfunding projects on network

platforms. Through empirical research, this study explores whether and how the potential variables in the model affect the public's donation willingness and behavior.

## Materials and methods

### Literature review

Network crowdfunding refers to a financing model supported by information technology (IT) in which entrepreneurs use online financing platforms to finance their products or creative ideas and public investors can invest a certain amount of money in return for donations or payment in kind [6–8].

Network crowdfunding, which is a relatively new phenomenon and an increasingly popular financing channel, combines modern social media technology and project-based financing. Scholars and practitioners agree that online crowdfunding generally has the potential to change traditional financing methods [9]. They have begun to analyze how new projects with business risks are successfully financed through crowdfunding [7,10] and what the influencing factors are that drive supporters to participate in crowdfunding projects.

Research on the influencing factors of charitable crowdfunding projects is relatively new and limited. Some foreign scholars have focused on the perspective of donors. Studies have confirmed that compared with commercial crowdfunding, charitable crowdfunding presents a unique situation [11], in that it relies more on the intrinsic value of the project and the social motivation of the donor, not the motivation of economic reward (external motivation). In particular, those who support nonprofit projects through crowdfunding are usually motivated by "sympathy and empathy for the matter, guilt for not paying, and continuously strengthened identity and social status". For donors, social identity and self-satisfaction with being an involved philanthropist constitute the main motivation for participation [12,13]. According to an empirical study by Hui et al. [14], multiple factors affect donor's participation in crowdfunding projects, including the consumption or experience of new products, the acquisition of spiritual or economic rewards, and social interaction. In addition, the empirical research of Shier and Handy [15] confirmed that the influence of others is a factor that affects people's willingness to donate online; furthermore, gender and perceptions of organizations are variables that affect the possibility of online donations. Sisco and Weber [16] analyzed the influencing factors of online donation behavior on the GoFundMe platform, and their survey found that women are more likely to resonate with donation information.

Some foreign scholars have conducted research from the perspective of network platform functions and capabilities. Wang and Fesenmaier [17] pointed out that the motivational factors of donation behavior in online interactions include tool effectiveness, quality assurance, status and expectations. Saxton and Wang [18] conducted empirical research on the nature and determinants of charitable donations in the social network environment and found that the success of fundraisers has nothing to do with the financial ability of organizations; rather, it is related to "network ability". The research of Sura et al. [19] also confirmed that the characteristics of Internet technology influence people's overall attitude towards online donation to a large extent and that their overall attitude positively affects people's willingness to donate through social networking sites.

Regarding research on the influencing factors of the public's willingness to donate on critical illness crowdfunding platforms, Chinese scholars mainly analyze critical illness crowdfunding platforms, help-seekers and donors perspectives.

Some studies from the perspective of critical illness crowdfunding platforms have confirmed that two factors, the project itself and the dissemination platform, have an impact on the fundraising effect of charitable projects carried out on a network platform [20–24]. On the

one hand, from the perspective of the project itself, the information presentation method (the framework effect and progress information) of public welfare crowdfunding projects affect the willingness of donors to participate in public welfare crowdfunding. At the same time, the target framework adjusts the facial expressions of help-seekers presented to donors. Influenced by empathy and willingness to donate, charitable crowdfunding donation behavior on online social platforms is positively affected by beneficiaries with happy facial expressions; that is, traditional "tear charity" has a limited effect in persuading people to donate, and donors are more willing to donate to charities that provide happy progress information. On the other hand, in terms of the communication platform, take the well-known WeChat platform as an example. People's WeChat circle is based on their real social circle. The scale, resource endowment and composition of individuals' real social circle affect their WeChat circle. As a result, the WeChat circle makes a major difference in the effect of crowdfunding [20,21]. In addition, Fan et al. [25] discussed the influence of the default option amount on personal donation willingness and its psychological mechanism in the context of online public welfare, and they found that a high-amount default setting and individuals' moral identity level have a moderating effect on donation willingness.

Some studies from the perspective of help-seekers have confirmed that the more diversified, the broader, and the larger the interpersonal network of help-seekers is, the better the fundraising effect [26]. The fundraising rate has a significant positive effect, and the higher the fundraising rate is, the greater the likelihood that financial transparency will be achieved [27].

Some studies from the perspective of donors have confirmed that young people are the main component of the online public welfare user community and that most participants participate in online public welfare activities by following public opinion leaders and, in the process, achieve identity recognition and generate social identity [28,29]. In the process of disseminating medical crowdfunding information, participants' behavior of forwarding medical crowdfunding information is mainly self-interested rather than altruistic. They are mainly relationship oriented and decide whether to forward such information according to the rules of favor exchange.

In summary, it is certain that the public's willingness to donate money and engage in donation behavior on critical illness crowdfunding platforms will be affected by the established technical functions of platforms, that is, the use of the existing platform system and the perspective of technology. In addition, the behavioral intentions of individuals will be affected by other internal and external factors.

## Theoretical basic research hypotheses

The TAM is selected to explore the trade-off process of the public when using system functions. As a general model, the TAM provides only two cognitive concepts that affect the willingness of individuals to accept technology, namely, perceived usefulness and perceived ease of use, and it does not interpret or limit the external variables that affect usefulness and ease of use for specific application situations [30]. Hence, the TPB is combined to incorporate variables, including public attitudes towards behavior, and the influence of others (social distance). On the basis of selecting the TAM and the TPB, to better explain the public's willingness to donate and decision-making behavior in the specific situation of a critical illness crowdfunding platform, public trust in the platform is regarded as an important premise of perceived usefulness, and the individual's empathic concern is added as variables to measure the public's willingness to donate to critical illness crowdfunding projects on network platforms.

**Technology acceptance model.**   Davis et al. [31] applied the theory of reasoned action (TRA) to the TAM in 1989; the model structure is shown in Fig 1. Technology acceptance

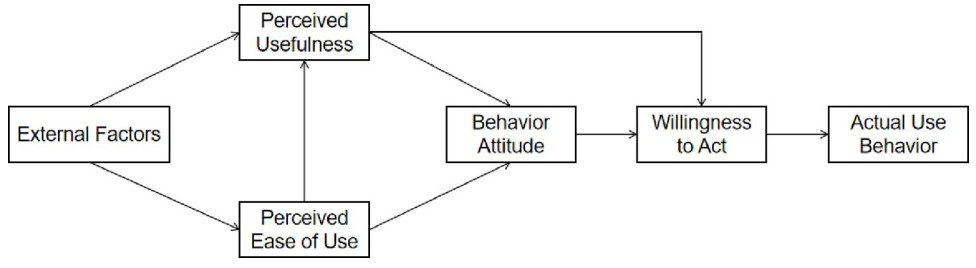

**Fig 1. Technology acceptance model.**

theory is used to study the influencing factors of individuals in accepting and using new information service systems. The TAM indicates that a user's actual use of a tool is directly determined by the will to use it. In addition, perceived ease of use and perceived usefulness are the main indicators of technology acceptance behavior, which in turn affect the user's behavioral attitude.

In the TAM, perceived ease of use refers to the ease of use of a tool by a user and is the process and effort expected. Perceived usefulness refers to the extent to which a user subjectively believes that the experience or efficiency brought by the use of a tool is enhanced and is the result and expected performance [31]. Perceived ease of use can also be used as a prerequisite for perceived usefulness, affecting the utility of a technology to the user and, in turn, the user's behavioral attitude [32]. In this study, donation attitude refers to the positive or negative attitude of the public towards donation behavior.

At a stage when information technology is not so developed, the higher the ease of use of a critical illness crowdfunding platform perceived by the public is, the higher the usefulness and effectiveness of the platform perceived by the public. However, the ease of use of the Internet platform has been very high after years of development, so the variable of the ease of use will not be considered in this study. Additionally, the spiritual satisfaction brought to the public by donations and the efficient experience of donation activities can positively affect the donation attitude of users. The public can complete donation activities through simple operations, and to meet the public's donation needs, critical illness crowdfunding platforms are extremely convenient. The more useful the platform, and the more positive the public attitude towards donation activities using the critical illness crowdfunding platform will be. Accordingly, this study proposes the following hypotheses:

Hypothesis 1 (H1). Perceived usefulness positively affects donation attitude.

**Interactivity and platform trust.** The rise of critical illness crowdfunding platforms and mobile payments has greatly increased the convenience of donation activities. However, because of the virtual, electronic and space-time extensibility of the platforms facing the public, the feedback involved in many traditional donation methods cannot be carried out, the information asymmetry between the two sides of the donation has increased, disadvantaged donors may encounter donation fraud, donors may be cheated due to the poor regulation of platforms, and control over the destination of the money donated is lacking. All of these factors will make donors feel that platform-based donation is riskier than traditional donation methods. Trust is considered one of the key factors in the success of crowdfunding projects since the emergence of charitable crowdfunding in the e-commerce era [33].

Since trust, as a simplified mechanism, can compensate for the inadequacy of reason and information and reduce complexity and uncertainty [34,35], MacMillan et al. [36] proposed

the concept of trust in contributions, arguing that trust is the key driver of commitment and that trust is built by shared values, communication and nonopportunistic behavior. In other words, donors must believe that they share values with the platform, and the platform must communicate its commitment to the destination of money and not behave in an opportunistic manner [36]. However, previous studies have shown that trust is essential for the existence and success of donor media and that trust is the foundation of charitable and voluntary organizations [37–40]. Therefore, this study adopts the view of MacMillan et al. [36], according to whom the trust of users is an important prerequisite for perceived usefulness.

Pavlou argued that inter-organizational trust is a form of institution-based trust, which embodies the secure feelings of institutions and structures, and comprises the structural assurance and situational normality of the Web [41]. Structural assurance highlights the reliance on protective structures such as guarantees, contracts, regulations, and transaction procedures to assure the expected outcomes. The structural assurance mechanism creates the belief that restrictive monitoring procedures associated with online transactions can effectively regulate the trading entities in terms of following transaction norms, thereby lessening uncertainty and opportunism. Tan and Thoen argued that the control mechanisms supplement party trust when information regarding the trading entities is insufficient or unavailable. Similarly, the empirical study conducted by Ratnasingamalso showed that trust in the technological infrastructure and online transaction environment are both essential for successful B2B transactions [42].

In other words, donors must believe that they share values with the platform, and the platform must communicate its commitment to the destination of money and not behave in an opportunistic manner [36]. However, previous studies have shown that trust is essential for the existence and success of donor media and that trust is the foundation of charitable and voluntary organizations [37–40]. Therefore, this study adopts the view of MacMillan et al. [36], according to whom the trust of users is an important prerequisite for perceived usefulness, and it believes that trust is very important for promoting the development of public donations using critical illness crowdfunding platforms, and Pavlou [43] & Donna (2006) [44] specifically proposed that trust affects perceived usefulness and perceived ease of use [45].

Furthermore, trust is always accompanied by certain risks. To ensure that the interests of trust will not be damaged, the existence of a trust guarantee mechanism is required. Trust can be divided into different types based on the guarantee mechanism. For example, Luhmann [46] divided trust into interpersonal trust and institutional trust. Interpersonal trust is based on familiarity and the relationship between humans and human feelings, while institutional trust reduces the complexity of social interaction through external, punitive or preventive mechanisms such as the law. Zucker [47] argued that trust can be divided into three levels: trust based on communication experience, i.e., the accumulation of interaction, exchange and trade experience; trust based on actors with sociocultural commonalities, i.e., obligations and cooperation rules of social imitation; and trust based on the system, i.e., nonindividual rules, social norms and institutions.

Based on the above analysis, the sources of donor trust in a platform can be summarized into two aspects: trust based on interaction and information exchange and trust based on platform rules. They are expressed as two aspects: interactivity and platform trust.

Interactivity refers to the public perception of the interactive services provided by a critical illness crowdfunding platform, which is based on the design of the interactive mechanism provided by the platform. The trust building process is inevitably accompanied by various social interactions and interpersonal behavior [47–49]. In the whole process of using a critical illness crowdfunding platform, if the platform can provide a complete donation experience to the public, from teaching the public how to use the platform to make donations to providing a

smooth mechanism of communication with other donors and help-seekers and having sufficient ability to solve the problems and doubts encountered by the public during use, the public may have a sense of trust in the platform. This trust is the public's confidence in the critical illness crowdfunding platform and is the expression of the exchange of emotional goodwill of both sides.

Hypothesis 2 (H2). Interactivity positively affects platform trust.

Hypothesis 3 (H3). Platform trust positively affects perceived usefulness.

**Empathy.** Empathy refers to an individual's understanding and response to the emotions of others. When an individual faces (or imagines) the emotional situation of one or more individuals, doing so first results in the sharing of emotions and feelings with others. Then, on the premise of the cognition of the difference between the self and others, the individual carries out a cognitive assessment of the overall situation, resulting in an emotional response accompanied by corresponding behavior (explicit or implicit behavior), and as the subject, the individual projects this emotion and behavior onto the psychological process of others as the object [50]. As a psychological phenomenon closely related to everyday life, empathy plays an important role in the process of personal socialization and interpersonal communication.

Kant said in his Critique of Practical Reason, "It is very good to do good to people out of love and sympathy, or to uphold justice out of love for order." Kant believes that compassion is an important internal driving force of charitable behavior. It inspires people to generate good intentions and charitable behavior. There is a natural feeling of sympathy in human nature. Individuals produce a mechanism of sympathy and empathy through compassion so as to resonate with others emotionally and cognitively. In this way, compassion opens up the emotional boundaries between individuals and others and has a sharing mechanism of social moral psychology [51].

Empathy is an important trait that affects donation behavior [52], which shows that donors with strong cognitive senses of other perspectives easily achieve emotional understanding and responses to the situations of others. They are more likely to show compassion and empathy for others, thus showing a stronger willingness to donate. Moreover, when the donation object is a specific unfortunate individual rather than a vague disaster group, it can stimulate the sympathy of donors, maximizing the empathy effect [53].

Donors are more likely to feel sad and to offer help when they believe that the recipients are similar to themselves [46]. The empathy dynamic model shows that when individuals face the emotion or situation of others, their own emotional cognitive system will be aroused. First, the sharing of emotion with others will be established. Then, individuals will realize the difference between themselves and others and think that their emotions are driven by others. Finally, their own high-level cognitions, such as moral norms and values, will be used to judge whether self-induced empathy with others is valid. If so, then the individual's cognition and the experience of emotion will combine to produce independent emotion and trigger the corresponding explicit or implicit behavior (or behavioral motivation). Finally, the individual will project his or her own cognitions and emotions onto others. The empathy dynamic model can be applied to the process experienced by the public when donating to critical illness crowdfunding donation projects on network platforms. Members of the public will see others dealing with misfortune and unfair circumstances. The higher the level of empathic concern of members of the public, the greater the likelihood that they will take the perspective of the help-seeker and form the same emotional consensus with the help-seeker.

In addition, according to Hoffman's empathy theory, when individuals see others in sadness, they will also feel sadness. After individuals help others, empathy sadness will dissipate.

Without help, empathy sadness will maintain a high level [54]. Therefore, a public with strong empathy ability is more likely to produce empathy sadness when they read on the internet about the sadness and difficulties of others. It is also more motivated to reduce sadness and obtain a positive experience through helping others.

Empathy has an important impact on the willingness of subjects to donate. It is regarded as a very important predictor of charitable donation behavior [55]. The research of stocks et al., shows that empathy is an important driving force of altruistic behavior and supports the empathy helping behavior hypothesis [56]. Many studies have shown that the stronger the empathy ability, the higher the willingness to donate [55,57,58].

Empathy largely depends on the automatic "activation" and "matching" state, producing common representations and similar emotions [59]. Nambisan and other scholars have pointed out that the expression of empathy is more common in online communities, especially in supportive online communities, which can promote online help [60]. The reason is that the overall atmosphere of the critical illnesscrowdfunding network platform, in addition to the common environment of empathic expression mentioned above, can also provide great conveniences for the completion of donation behavior, freeing donors in terms of implementation time and form [61]. It can be said that donor empathy cognition is more easily stimulated in the emotionally expressive atmosphere of serious illness crowdfunding. Donors will also perceive that such platforms can complete the process from stimulation of donor empathy to donation behavior. Not only will doing so enhance the recognition of platform usefulness, but it will also increase donation behavioral attitude through the network platform, that is, perceived platform usefulness. Additionally, the higher the level of empathic concern of individuals is, the more significant their contributions will be. Accordingly, this study proposes the following hypotheses:

Hypothesis 4 (H4). Empathic concern positively affects perceived usefulness.

Hypothesis 5 (H5). Empathic concern positively affects donation attitude.

**Theory of planned behavior and Social distance.** The TPB holds that the behavior of an individual is not always controlled by his or her own will [62]. Rather, the individual's behavior is affected by internal and external factors. Attitudes and standardized responses to perceived situations will change the individual's behavior. The TPB mainly considers the attitude and perceived behavioral control of individuals based on their own interests and the rational choices they make in regard to the necessity of engaging in a particular behavior [63].

In the process experienced by the public when making donations by using critical illness crowdfunding platforms, donation attitude mainly refers to the positive or negative attitude of the public towards donation behavior, that is, the subjective feelings of the public towards donation behavior. The more positive the public's attitude towards donation using critical illness crowdfunding platforms is, the higher the willingness to use such platforms to donate money. In recent years, many scholars have integrated TAM and TPB. For example, Luo and other scholars have used it to study people's willingness to use *yu'e Bao* [64]. Chen et al.have built a research model on the influencing factors of public welfare crowdfunding donation willingness based on WeChat social media [65]. The empirical results also directly verify that individual donation attitudes directly impact donation intention. Feng analyzed the influencing factors of platform user willingness to donate by 802 questionnaires, and the results showed that user donation attitudes and perceived behavioral controls are the most influential [66].

Although traditional studies on the integration of TAM and TPB generally consider subjective norms as an important variable, Subjective norms refer to the social pressure that an

individual feels when engaging in a certain behavior, which is manifested in the individual's perception of the influence of friends, family and society on whether or not to make a donation. In this study, we use social distance as a variable to measure the influence of social pressure on donation behavior. Because, as far as this study is concerned, users' behavior of participating in crowdfunding for serious diseases and their related feelings cannot be simply summarized as herd psychology. The influence of social relations on users, in addition to the relatively important individual influence, will also be affected by the clustering effect of certain scale group participation or communication under the corresponding circumstances, which can be understood as the group appeal and influence from social relations [60].

The concept of social distance, which originated in sociology, refers to the degree of closeness existing between groups and individuals, and the degree of closeness measures the influence that one party has on another [67]. Social distance reflects the characteristics of psychological distance between the perceived self and others or groups, which in turn describes the degree of intimacy between them, such as the similarity between the self and others. Identity is an important factor that affects the distance between individuals and society [68]. With the development of the new media, the network environment has undergone significant changes. It provides us with increasing functions of instant communication, which also changes the psychological judgment of social distance between people and the surrounding environment. Thus, the behavior of individuals in the network environment will naturally be different. If media information dissemination is taken as the starting point, social distance mainly exists between the information communicator and the information receiver, and the distance between the two sides will affect the establishment of the communication and reception relationship [69]. As the network of critical illnesscrowdfunding breaks the geographical and social distance of traditional offline donations, there may be a strong, weak, or even negligible social relationship between donors and the recipient(s). Therefore, in the network crowdfunding, the relationship type, where donor and recipient are familiar with each other and have strong interaction and dependence, is defined as the *close social distance*, and the corresponding alienated relationship, with less understanding, is defined as *no interactive relationship distance* [69].

In the dimension of interactive environment, the above social relations mainly reflect the impact of the mutual relationship between donors, helpers, forwards and users on users' participation in public welfare crowdfunding [24]. Understood through the concept of social distance, they reflect the closeness or clarity of the relationship between individuals. Previous studies have confirmed that social distance and intimacy have a significant impact on user emotions and perceptions, mainly reflected in the two initial concepts of peer influence and clustering effect. Among them, peer influence is the more significant factor. Under the influence of social relations, potential donors trust donation initiatives from "peers" [25]. In addition, this peer influence lies not only in the strengthening of trust but also has a significant impact on the stimulation of user compassion, the amount and frequency of user donations, and other costs that users are willing to pay.

According to the cognitive level theory (CLT) proposed by Liberman and Trope [70], people's behavior and cognition will be different due to the change of psychological distance [71]. This theoretical framework supports a large number of studies of the internal psychological mechanism behind individual behaviors from the perspective of psychosocial distance [72]. Fruitful results have been achieved in this field. Some scholars point out that the public is more inclined to donate to people and organizations they know [73]. Research by Ein-Gar and Levontin (2013) demonstrates how individual charitable donations increase with the reduction of social distance. This result shows that charitable donations will be affected by the donation relationship. The closer the relationship, the more individual donations [74]. Miao Qing has

noted that the shorter the social distance, the greater the possibility of arousing compassion and the willingness to donate to each other [75]. Earlier studies by Small and Simonsohn pointed out that people will have sympathy for victims who are close, and that this sympathy can be extended to other victims with similar experiences, so as to increase the willingness of individuals to help these victims. For example, if an individual family member suffers from breast cancer, the individual will be more sympathetic to a group with breast cancer and more willing to contribute to that group. This transmission effect is stronger when individuals are more closely related to the victims [76].

There were some experimental studies that demonstrate the effect of social distance on donation behavior. Rachlin and Jones [77] divided subjects into an ingroup and an outgroup. They found that in the ingroup group, the psychological distance between individuals was closer and that the behavior of helping each other appeared more frequently. In contrast, in the outgroup, the psychological distance from people in the outgroup was farther, and individuals were not willing to help them. In a study by Stephan et al. [78], who took a temporal perspective, the authors found that if the description of an individual is more abstract, subjects will perceive a farther social distance; that is, they will be more unwilling to help him or her, and they will assign fewer tokens to him or her. In contrast, when the description of an individual is more specific and detailed, subjects will make the judgment that the social distance is closer, and they will be more willing to help him or her and assign more tokens to him or her [78]. That is, if there is a larger social distance between a donor and a helper-seeker, helping behavior will be less likely to occur, and the same result applies in the case of spatial distance [79]. Regarding the length of time to read online help-seeking information, the main effect of social distance is significant. In the face of the same content of help-seeking information, information from good friends needs a shorter reading response time than that from strangers, and people will be more trusting of information with a close social distance [80]. Based on the above factors, this study proposes the following hypotheses:

Hypothesis 6 (H6). Donation attitude positively affects donation willingness.

Hypothesis 7 (H7). Donation willingness positively affects donation behavior.

Hypothesis 8 (H8). Social distance positively affects donation behavior.

Based on the above assumptions, we obtain a model of the influencing factors of the public's donation behavior of using critical illness crowdfunding platforms. The model is shown in Fig 2.

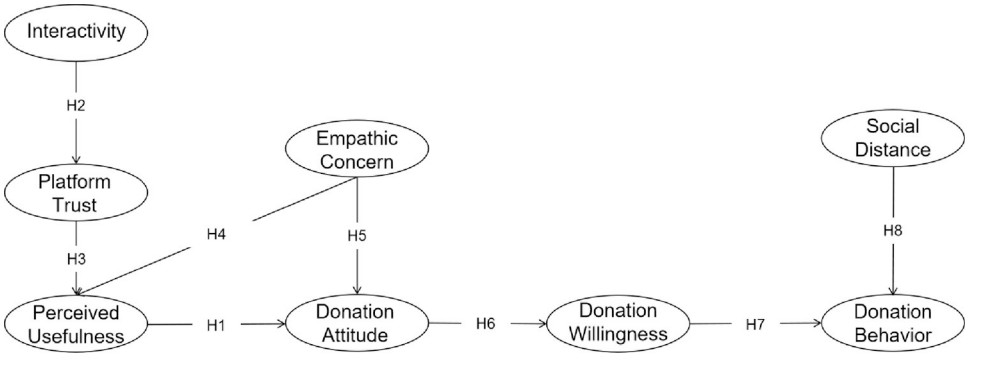

**Fig 2. Diagram of the proposed model.**

## Methodology

Structural equation modeling (SEM) is an advanced statistical method developed on the basis of factor.Analysis and path analysis. In 1918, Fisher et al. [81] proposed path analysis in genetics, and introduced the path diagram and obtained the basic form of SEM. Joreskog et al. [82] put forward the preliminary concept of SEM in the early 1970s, and divided SEM into structural model and measurement model. Since then, SEM has been widely used in biology, medicine, education, behavior, psychology and many other fields. In the field of sociology, Yang W et al. [83] used SEM to evaluate the social impact of construction projects and the relationship between them and public response, and to identify and resolve conflicts from the perspective of social risk management.

**Survey design.**    The studies involving human participants were reviewed and approved by the studies involving human participants were reviewed and approved by the Ethics Committee (HREC) of the School of Economics and Management in East China Jiaotong University. The participants provided their written informed consent to participate in this study. The questionnaire consisted of two parts. The first part collected data on demographic variables, which were used as control variables: gender, age, education, occupation, and income. The second part was the measurement scale. To ensure the reliability and validity of the measurement, this study selected mature scales from home and abroad and modified the wording based on the actual needs of this study.

A total of 8 latent variables and 31 measurement variables were included in the measurement scale. All scales were scored on a 5-point Likert scale ranging from 1 ("completely disagree") to 5 ("completely agree"). The factor loadings and descriptive statistics of each measurement variable, which were obtained by SPSS 22.0, are shown in Table 1.

1. The donation attitude scale refers to Bagozzi's et al. [84] definition of behavioral attitude measurement in the TPB. The items of donation attitude measurement are divided into three dimensions: "what is good", "what is worth doing" and "what am I willing to do".

2. The empathy scale is based on the Interpersonal Reactivity Index (IRI) proposed by Davis [85], which has been proven to be suitable for different normal population groups. The IRI is an instrument for measuring empathic concern based on a multidimensional theoretical framework of empathy [85,86]. Empathy is divided into four dimensions: perspective selection (PT), empathetic concern (EC), imagination (FS), and personal pain (PD). Among them, EC is mainly used to measure the degree of individuals' emotional care, warmth and sympathy for others.

3. Perceived usefulness is used to measure whether a new good or service can improve the original work efficiency. In the TAM, Davis [31] pointed out that the more obviously a good or service can improve the original work efficiency, the more consumers will perceive the value of the good or service, which in turn will affect consumers' purchase decision. In general, perceived usefulness is assessed from the perspective of convenience and contrast (i.e., compared with the original or other products). This study examines the perceived usefulness of user contributions to critical illness crowdfunding platforms from the perspective of convenience, contrast, and spiritual satisfaction.

4. The platform trust scale is derived from the trust scale proposed by McKnight et al. [87]. There are 6 original items. Notably, in the pretest, we found that some items between platform trust and interactivity were highly correlated; thus, we eliminated the redundant measurement variables, such as "I believe critical illness crowdfunding platforms have sufficient capacity to effectively deal with the various situations encountered by users in donation

**Table 1. Factor loadings, descriptive statistics and scale sources of each measured variable.**

| Potential variables | Measured variables | Item | Mean | SD | Source |
|---|---|---|---|---|---|
| Donation attitude | Donation through critical illness crowdfunding platforms is something that can help people in need solve problems | A1 | 3.50 | 1.23 | [80] |
| | Donations through critical illness crowdfunding platforms are worth doing | A2 | 3.61 | 1.16 | |
| | Donation through critical illness crowdfunding platforms is something that you would love to do | A3 | 3.55 | 1.11 | |
| Empathic Concern | I feel soft and caring when I see those less fortunate than me on critical illness crowdfunding platforms | B1 | 3.59 | 1.19 | [85,86] |
| | I want to protect people from being used by others on critical illness crowdfunding platforms | B2 | 3.55 | 1.16 | |
| | I feel very sympathetic when I see people being treated unfairly on critical illness crowdfunding platforms | B3 | 3.71 | 1.06 | |
| | I feel sorry when I see someone in trouble or with a problem on critical illness crowdfunding platforms | B4 | 3.70 | 1.15 | |
| | I think that I am a very soft-hearted person | B5 | 3.51 | 1.13 | |
| Perceived usefulness | Donations can be made simple through critical illness crowdfunding platforms | C1 | 3.74 | 1.09 | [31] |
| | Through critical illness crowdfunding platforms, donation efficiency can be improved | C2 | 3.72 | 1.07 | |
| | I can have a satisfying donation experience through critical illness crowdfunding platforms | C3 | 3.65 | 1.13 | |
| | Giving money through critical illness crowdfunding platforms is spiritually satisfying | C4 | 3.60 | 1.11 | |
| Platform trust | I believe that critical illness crowdfunding platforms put the interests of users first | D1 | 3.51 | 1.14 | [87] |
| | I believe that critical illness crowdfunding platforms are very concerned about the interests of users, not just their own interests | D2 | 3.50 | 1.14 | |
| | I think that critical illness crowdfunding platforms are credible | D3 | 3.52 | 1.17 | |
| | I believe that critical illness crowdfunding platforms will meet their responsibilities and serve people in need | D4 | 3.47 | 1.17 | |
| Interactivity | Critical illness crowdfunding platforms provide a smooth platform for information exchange and communication between help-seekers and donors | E1 | 3.51 | 1.15 | [88] |
| | Through critical illness crowdfunding platforms, donors can easily communicate with other donors | E2 | 3.64 | 1.12 | |
| | Critical illness crowdfunding platforms encourage users to familiarize themselves with the platform and actively participate in and use it | E3 | 3.64 | 1.14 | |
| | Through critical illness crowdfunding platforms, donors can receive enough feedback from fundraisers (message responses, e-mail acknowledgments, etc.) | E4 | 3.60 | 1.05 | |
| | For all kinds of questions and issues raised by users, critical illness crowdfunding platforms can solve problems and answer quickly and effectively | E5 | 3.61 | 1.15 | |
| Donation willingness | In the future, I will try to participate in donating on critical illness crowdfunding platforms | F1 | 3.66 | 1.15 | [90] |
| | In the future, I will make donations on critical illness crowdfunding platforms as a way to practice the public good | F2 | 3.65 | 1.16 | |
| | In the future, I will recommend the use of critical illness crowdfunding platforms to make donations | F3 | 3.63 | 1.15 | |
| Social distance | I prefer to donate money to relatives | G1 | 3.54 | 1.13 | [92] |
| | I prefer to donate money to friends, including the relatives of friends | G2 | 3.63 | 1.14 | |
| | I prefer to donate money to colleagues and classmates, including the relatives of colleagues and classmates | G3 | 3.56 | 1.19 | |
| | I am more likely to donate money to acquaintances, including the relatives of acquaintances | G4 | 3.57 | 1.14 | |
| | I prefer to donate money to strangers | G5 | 3.32 | 1.16 | |
| Donation behavior | How much money have I donated to acquaintances over the past year? | H1 | 3.61 | 1.15 | [93] |
| | How much money have I donated to strangers over the past year? | H2 | 3.34 | 1.09 | |

activities" and "I believe that if there is a problem during the donation process, critical illness crowdfunding platforms will help users appropriately solve it". Platform trust was reduced from 6 questions to 4 questions.

5. The website interactivity scale draws on the communication items in the perceived interactivity scale developed by McMillan and Hwang [88]. In this study, the interactivity of donation platforms refers to the design of platform-based donations perceived by users to meet the various communication needs of users. It is a measure of the degree of satisfaction with platform-based donations with regard to meeting users' needs to participate in interaction, exchange and obtain feedback.

6. The scale of donation willingness mainly refers to a relevant consumer behavior intention-related scale by Shier and Handy [89]. The three dimensions are measured based on the possibility of participation, the possibility of repeated participation and willingness to recommend, for a total of three items [90].

7. The social distance scale was designed by Bogardus [91] and has repeatedly been used in the Bogardus Social Distance Scale, with Lee et al. [92] making appropriate modifications to the scale. In this study, the measurement of social distance is divided into five items: relatives, friends (including the relatives of friends), colleagues/classmates (including the colleagues and relatives of classmates), acquaintances (including the relatives of acquaintances), and strangers.

8. The donation behavior scale refers to Zhang [93] study on the influencing factors of the charitable donation behavior of Chinese urban residents. In this study, donations to acquaintances and strangers are used as the measurement classification for user platform contributions.

**Data collection and samples.**   This study was conducted in the form of a questionnaire survey, including field research and network research. The survey was conducted from February 2020 to April 2020. The survey completion process involved maintaining communication with the respondents. At the beginning of this process, the questionnaire indicated that the survey was for research purposes only. The survey involved a nationwide scale, with 1,000 questionnaires sent and 768 questionnaires returned. After removing invalid questionnaires with obvious regularities, short completion times and inconsistent answers, 710 valid questionnaires were retained, and the effective response rate reached 92.4%.

The questionnaire respondents consisted of 359 males (50.6%) and 351 females (49.4%); the ratio of men to women was relatively even. Employees aged between 18 and 40 accounted for 78.4%. Those with a bachelor degree accounted for 40.7% of the sample, followed by those with a college degree (29.7%); the number of respondents with master's or doctoral degrees and the number of respondents with a high school (secondary school) educational level and below were both approximately 15%. In addition to full-time students, there were relatively high proportions of sales personnel (14.5%), production personnel (9.9%) and technical/research and development (R&D) personnel (8%); employees in remaining industries, such as marketing/public relations personnel, human resources personnel, teachers, and financial/audit personnel, accounted for approximately 4%. Regarding the distribution of respondents by income, those earning 5,000–10,000 yuan per month accounted for the majority (33.2%), followed by those earning less than 2,000 yuan per month (24.9%). The proportion of respondents in the total sample with a monthly income of 2,000–5,000 yuan and more than 10,000 is approximately 20%.

We also found that regarding information on critical illness crowdfunding platforms, 60.1% of the respondents said that they were recommended. Additionally, 49.3% said that they were informed by doctors/volunteers in hospitals, and 42.5% indicated that they had been informed after searching for information.

## Data analysis

**Exploratory factor analysis.**   The primary purpose of exploratory factor analysis is to determine the number of observed variables. An exploratory analysis of 710 valid samples was performed before formal factor analysis.

At this stage, the reliability and validity tests are used to test the reliability of the data and the consistency of the question. The multivariate normality of the data is determined by describing the skewness and kurtosis of the observed variables. After meeting the above requirements, the principal component analysis method was used for factor analysis and eight comprehensive factors were extracted. Through skew rotation, they reflect the donation behavior and other factors. The results provide a basis for establishing the hypothesis of confirmatory factor analysis.

**Reliability and validity tests.** Reliability is a measure of the stability and internal consistency of the results measured by a scale. This study mainly adopts the internal consistency coefficient (Cronbach's α) and composite reliability (CR) as test indicators. Donation behavior had a low Cronbach's α, but its CR value was 0.811 (which is still acceptable). The Cronbach's α coefficients and CR values of the remaining variables were higher than 0.7, indicating that the scale as a whole had better reliability (in Table 2).

**Validity test.** Bartlett's sphere and KMO tests were performed using SPSS 22.0. The results in Table 2 show that the overall KMO value of the questionnaire is 0.978 and the KMO

**Table 2. Reliability and validity index coefficients.**

| Potential variables | Item | AVE | CR | Cronbach's α | KMO |
|---|---|---|---|---|---|
| Donation attitude | A1 | 0.6797 | 0.8642 | 0.764 | 0.694 |
|  | A2 |  |  |  |  |
|  | A3 |  |  |  |  |
| Empathic concern | B1 | 0.6086 | 0.8857 | 0.838 | 0.858 |
|  | B2 |  |  |  |  |
|  | B3 |  |  |  |  |
|  | B4 |  |  |  |  |
|  | B5 |  |  |  |  |
| Perceived usefulness | C1 | 0.5835 | 0.8484 | 0.761 | 0.772 |
|  | C2 |  |  |  |  |
|  | C3 |  |  |  |  |
|  | C4 |  |  |  |  |
| Platform trust | D1 | 0.6622 | 0.8869 | 0.830 | 0.813 |
|  | D2 |  |  |  |  |
|  | D3 |  |  |  |  |
|  | D4 |  |  |  |  |
| Interactivity | E1 | 0.5654 | 0.8667 | 0.807 | 0.843 |
|  | E2 |  |  |  |  |
|  | E3 |  |  |  |  |
|  | E4 |  |  |  |  |
|  | E5 |  |  |  |  |
| Donation willingness | F1 | 0.6719 | 0.8600 | 0.756 | 0.693 |
|  | F2 |  |  |  |  |
|  | F3 |  |  |  |  |
| Social distance | G1 | 0.5916 | 0.8785 | 0.827 | 0.854 |
|  | G2 |  |  |  |  |
|  | G3 |  |  |  |  |
|  | G4 |  |  |  |  |
|  | G5 |  |  |  |  |
| Donation behavior | H1 | 0.6823 | 0.8111 | 0.535 | 0.600 |
|  | H2 |  |  |  |  |

**Table 3. The values of skewness and kurtosis.**

|  | A1 | A2 | A3 | B1 | B2 | B3 | B4 | B5 | C1 | C2 | C3 | C4 |
|---|---|---|---|---|---|---|---|---|---|---|---|---|
| Skewness | 0.393 | -0.515 | -0.434 | -0.450 | -0.384 | -0.492 | -0.581 | -0.465 | -0.529 | -0.688 | -0.455 | -0.488 |
| Kurtosis | 0.863 | 0.563 | 0.507 | -0.752 | -0.820 | -0.556 | -0.471 | -0.486 | -0.563 | -0.095 | -0.725 | -0.502 |
|  | D1 | D2 | D3 | D4 | E1 | E2 | E3 | E4 | E5 | F1 | F2 | F3 |
| Skewness | -0.401 | -0.451 | -0.439 | -0.401 | -0.464 | -0.495 | -0.477 | -0.493 | -0.455 | -0.531 | -0.578 | -0.564 |
| Kurtosis | -0.723 | -0.461 | -0.633 | -0.623 | -0.484 | -0.615 | -0.656 | -0.334 | -0.761 | -0.622 | -0.573 | -0.498 |
|  | G1 | G2 | G3 | G4 | G5 | H1 | H2 |  |  |  |  |  |
| Skewness | -0.427 | -0.400 | -0.459 | -0.472 | -0.261 | -0.468 | -0.059 |  |  |  |  |  |
| Kurtosis | -0.583 | -0.776 | -0.681 | -0.566 | -0.702 | -0.614 | -0.921 |  |  |  |  |  |

values of the eight latent variables are greater than 0.5. The Sig value of the Bartlett sphere test is 0, which is less than 0.01, indicating that the sample data is highly correlated. The above results show that the sample data have good convergence validity.

Skewness and kurtosis can effectively describe the normal distribution of data for each observed variable [94]. The values of skewness and kurtosis are shown in Table 3 within ±1 [95,96]. Absolute value of each skewness is less than 3, and absolute value of each kurtosis is less than 10 [97]. All variables and averages are in accordance with the standard, which indicates the data show a normal distribution.

**Factor analysis.** Factor analysis represents the basic structure of data by analyzing the internal mapping between variables. In this study, the principal components were analyzed by SPSS 22.0. Factor analysis can group variables according to mapping degree, extract key indicators of different groups, and calculate cumulative variance contribution rate. These key indicators directly reflect the basic structure of things. As can be seen from Table 4, the exploratory factor loads of 31 indicators were obtained through the maximum variance rotation analysis. These quantities are all greater than 0.5, satisfying the model requirements and allowing the extraction of 8 factors to explain the structure of the variables. After tilting and rotating, the latent variable explained much more than 50% of the initial information, which fully reflected the influencing factors of donation behavior.

## Confirmatory factor analysis

CFA is performed to confirm the quality and adequacy of the measurement model. When using normally distributed data, ML is suitable to estimate the model [98]. The correlation coefficient matrix is shown in Table 5. The correlation values of all observed variables ranged from 0.425 to 0.803, indicating a significant correlation. The initial model establishment is

**Table 4. Rotated factor analysis(Factor analysis).**

| Donation attitude | | Empathic concern | | Perceived usefulness | | Platform trust | | Interactivity | | Donation willingness | | Social distance | | Donation behavior | |
|---|---|---|---|---|---|---|---|---|---|---|---|---|---|---|---|
| V1 | Factor | V2 | Factor | V3 | Factor | V4 | Factor | V5 | Factor | V6 | Factor | V6 | Factor | V7 | Factor |
| A1 | 0.838 | B1 | 0.822 | C1 | 0.783 | D1 | 0.811 | E1 | 0.772 | F1 | 0.808 | G1 | 0.759 | H1 | 0.826 |
| A2 | 0.829 | B2 | 0.705 | C2 | 0.793 | D2 | 0.804 | E2 | 0.748 | F2 | 0.823 | G2 | 0.784 | H2 | 0.826 |
| A3 | 0.806 | B3 | 0.762 | C3 | 0.764 | D3 | 0.816 | E3 | 0.767 | F3 | 0.828 | G3 | 0.787 |  |  |
|  |  | B4 | 0.807 | C4 | 0.713 | D4 | 0.824 | E4 | 0.759 |  |  | G4 | 0.796 |  |  |
|  |  | B5 | 0.799 |  |  |  |  | E5 | 0.712 |  |  | G5 | 0.717 |  |  |
| Cumulative variance interpretation % | | | | | | | | | | | | | | | |
| 67.939 | | 60.874 | | 58.348 | | 66.205 | | 56.493 | | 67.238 | | 59.138 | | 68.271 | |

**Table 5. Correlation coefficient matrix and the square root of the AVE.**

| | 1 | 2 | 3 | 4 | 5 | 6 | 7 | 8 |
|---|---|---|---|---|---|---|---|---|
| 1. Donation attitude | 0.824 | | | | | | | |
| 2. Empathic concern | 0.720** | 0.780 | | | | | | |
| 3.Perceived sefulness | 0.621** | 0.643** | 0.764 | | | | | |
| 4. Platform trust | 0.537** | 0.500** | 0.585** | 0.814 | | | | |
| 5. Interactivity | 0.553** | 0.572** | 0.644** | 0.775** | 0.752 | | | |
| 6.Donation willingness | 0.766** | 0.722** | 0.635** | 0.512** | 0.555** | 0.820 | | |
| 7. Social distance | 0.804** | 0.749** | 0.658** | 0.555** | 0.598** | 0.803** | 0.769 | |
| 8. Donation behavior | 0.587** | 0.567** | 0.469** | 0.425** | 0.454** | 0.625** | 0.633** | 0.826 |

**. Expresses that the correlation is significant at the 0.01 level (two-tailed).

shown in Fig 3. Before assessing the fitness of the model, it is necessary to first test the "Offending Estimate" to determine whether the estimated coefficient is within the acceptable range. Hair et al. [99] believes that the offending estimate follows three rules: First, there is a negative

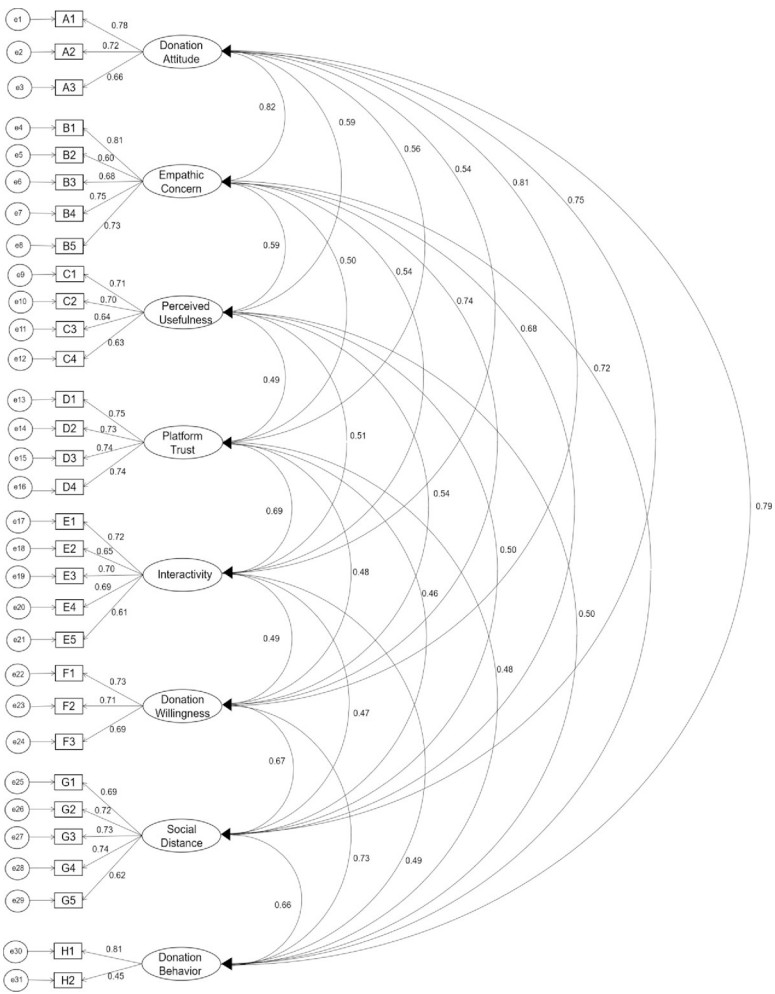

**Fig 3. Confirmatory factor analysis.**

**Table 6. Fit index calculations.**

| Fit indexes | Absolute fit indexes | | | | Relative fit indexes | | |
|---|---|---|---|---|---|---|---|
| | RMSEA | GFI | $x^2$/df | AGFI | CFI | NNFI | NFI |
| CFA | 0.027 | 0.948 | 1.501 | 0.937 | 0.981 | 0.979 | 0.946 |
| SEM | 0.046 | 0.916 | 2.524 | 0.901 | 0.940 | 0.935 | 0.905 |
| Suggested value | <0.08 | >0.9 | <3 | >0.9 | >0.9 | >0.9 | >0.9 |

error term variation. Second, the standardization coefficient is too close to or exceeds 1 (generally 0.95 is the threshold value); in addition, the standard error is too large. If the test results do not have the above offending estimation characteristics, the model is qualified in the preliminary test and can be tested for fitness.

To evaluate the internal structure fit, average variance extracted (AVE) can be used to assess the significance of the estimated parameters in the model, the indices and the reliability of the latent variables. In addition, composite reliability (CR) is used to evaluate the consistency of the measured variables. The CR should be greater than or equal to 0.6 and the AVE test should be greater than 0.5 to meet the intrinsic quality verification analysis standard of the model.

As shown in Table 1, the factor loadings of all measurement items were between 0.705 and 0.838, and in Table 2 the CR values of the variables were higher than 0.7, indicating that the scale as a whole had better reliability. As shown in Table 2, the AVE values of all latent variables were between 0.5654 and 0.6823, meeting requirements and indicating that the questionnaire as a whole has good convergent validity. Discriminant validity is tested by comparing the arithmetic square root of the AVE of each latent variable with the correlation coefficient of that variable and other variables to verify discriminant validity. When the arithmetic square root of the AVE is higher than the correlation coefficient of the two variables, the scale has good discriminant validity. As shown in Table 4, the arithmetic square root of the AVE of each variable is larger than the correlation coefficient of the two variables, indicating that the questionnaire as a whole has good discriminant validity.

AMOS 21.0 was used to test the hypotheses of this study, the distributions of potential variables and determined have been observed, that they are all approximately "normality". In this special case, variables are classified, SEM can only be used approximately. Table 6 shows the fit index calculations, where $x^2$/df is 1.501 (<3), RMSEA is 0.027 (<0.08), GFI is 0.981 (>0.9), NFI is 0.946 (>0.9), CFI is 0.981 (>0.9), NNFI is 0.979 (>0.9), AGFI is 0.937(>0.9). The measurement model appears to be acceptable, as the above indices are in line with the evaluation criteria, indicating the model-fit is good.

**Test of the SEM constructs and correlations.** A new SEM is developed to examine the relationships between the eight influencing factors to determine how to promote the donation behavior in the future. From Fig 4, the relationships between the independent and dependent variables can be measured by a path diagram. As shown in Table 6, the same indices can be used to evaluate adequacy of both the CFA and SEM. All indices are in the appropriate range, indicating the model is acceptable. All hypotheses are supported by the data in Table 7. Based on the path analysis results, H1-H8 all pass the test.

## Results

Based on the TPB, this study proposes and verifies that the public's social distance and willingness to donate to critical illness crowdfunding projects on online platforms are variables that positively affect the public's behavior of donating to critical illness crowdfunding projects. Additionally, the effect of the social distance variable is verified, and the dissemination of help-

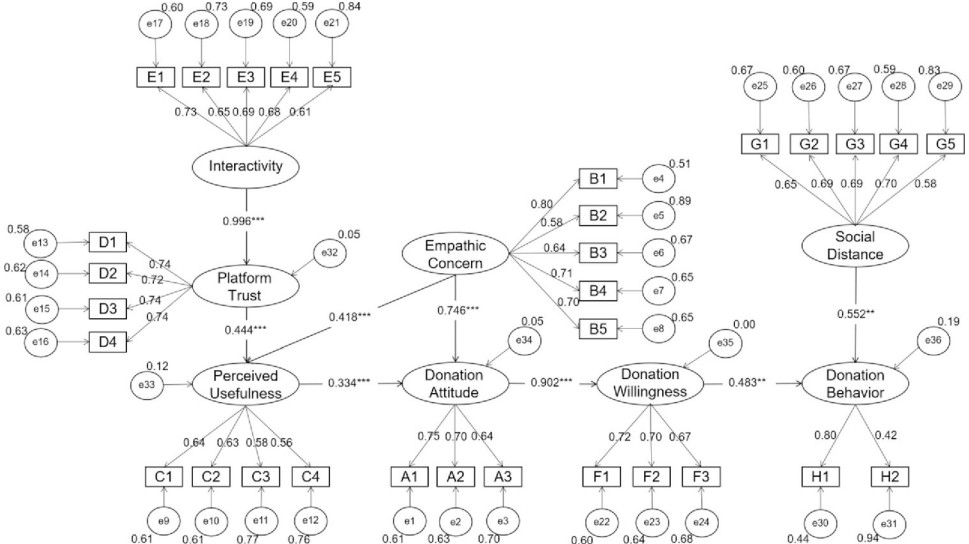

**Fig 4. Path coefficients of the hypothetical model.** Note: Significance levels * p<0.05,** p<0.01,*** p<0.001.

seeking information is affected by the habitual thinking in the "social acquaintance" environment. In the process of information dissemination and reception, people who have demographic factors similar to those of recipients or people who are similar to recipients and have the same values are often psychologically positioned as "owners" and have strong emotional resonance, which triggers donation behavior. In addition, donation attitude positively affects donation behavior through donation willingness.

At the same time, this study verifies that empathic concern significantly positively affects perceived usefulness and that interactivity positively affects platform trust, which in turn affects perceived usefulness. When the public believes that the interactions and information exchanged on online critical illness crowdfunding platforms are credible, this credibility will increase their trust in such platforms, and they will then agree more with the view that "donating through the critical illness crowdfunding network platforms can help the target audience", enhancing the public's willingness to donate and its actual donation behavior.

**Table 7. Structural model results.**

| Hypothesis | Proposed Effect | Path Coefficient | S.E. | C.R. | P | Results |
|---|---|---|---|---|---|---|
| H1: Perceived usefulness positively affects donation attitude. | + | 0.334 | 0.050 | 6.660 | *** | H1 is supported |
| H2: Interactivity positively affects platform trust. | + | 0.996 | 0.055 | 18.073 | *** | H2 is supported |
| H3: Platform trust positively affects perceived usefulness. | + | 0.444 | 0.036 | 12.179 | *** | H3 is supported |
| H4: Empathic concern positively affects perceived usefulness. | + | 0.418 | 0.032 | 12.935 | *** | H4 is supported |
| H5: Empathic concern positively affects donation attitude. | + | 0.746 | 0.043 | 17.317 | *** | H5 is supported |
| H6: Donation attitude positively affects donate willingness. | + | 0.902 | 0.047 | 19.150 | *** | H6 is supported |
| H7: Donation willingness positively affects donation behavior. | + | 0.483 | 0.186 | 2.604 | 0.009 | H7 is supported |
| H8: Social distance positively affects donation behavior. | + | 0.552 | 0.213 | 2.598 | 0.009 | H8 is supported |

Note: Significance levels

* p<0.05

** p<0.01

*** p<0.001.

Based on the TAM and the TPB, this study verifies perceived usefulness and empathic concern positively affect the public's attitude towards donations, which in turn affects the public's donation willingness. Through simple and convenient operation and the lack of restrictions due to time and space, the public feels that it is easy to donate money on online platforms, and it can quickly complete the whole process on a platform and obtain spiritual satisfaction. The perceived usefulness and efficiency of platforms have a positive impact on the public's attitudes towards using critical illness crowdfunding platforms to make donations.

## Discussion

### The theoretical contribution of the study

**Integrating the TAM and TPB and systematically studying the influencing factors of donations for critical illness crowdfunding projects on network platforms.** Regarding research on the influencing factors of the public's willingness to donate on critical illness crowdfunding platforms, most Chinese scholars conduct analysis based on a single dimension. This study is based on stakeholder theory, social identity theory, and the integrated TAM and TPB. It analyzes critical illness crowdfunding platforms, help-seekers and donors as a whole and systematically proposes the influencing factors of the public's willingness to donate to critical illness crowdfunding projects on network platforms. Regarding the factor model, our empirical research clearly and comprehensively shows the relationships between different variables and the effect of each variable on donation willingness and donation behavior, and it provides a theoretical basis for how to improve the public's willingness to donate to critical illness crowdfunding projects.

**At the theoretical and practical level, enriching research on the TAM and TPB.** The TAM is used to study the factors that affect individuals in the process of accepting and using new information service systems. The TAM points out that the actual use behavior of a certain tool is directly determined by the user's willingness to use it. In addition, perceived usefulness are the main indicators of technology acceptance, which in turn affect users' behavior and attitude. (1) This study innovatively introduces two variables, empathic concern and platform trust, further enriching theoretical and practical research on the TAM. Based on the results of this study, when donors have feelings of empathy for help-seekers, they are more likely to have the emotion of sadness. This emotion will make donors more inclined to provide donations; that is, empathic concern positively affects donation attitude [52,55,57,58], and platform trust also affects perceived usefulness [36,43–45]. This study has laid a more solid theoretical foundation for research on the TAM and on the factors that influence the public's willingness to donate to critical illness crowdfunding projects on network platforms. (2)This study's innovative introduction of the variable social distance further enriches theoretical and practical research on the TPB. The TPB believes that an individual's behavior is not always controlled by his or her own will. Rather, it is also due to internal and external factors that affect the individual's expected behavior. This study finds that members of the public are more inclined to give to people and organizations whom they are familiar with. That is, social distance positively affects donation behavior [73,75]. This study enriches the theoretical basis of research on the factors that influence the public's willingness to donate to critical illness crowdfunding projects on network platforms.

**Exploring the path of influence from trust to public donation behavior.** Unlike previous studies, which have focused on the direct relationship between trust and donation willingness, this study shows that the more the public trusts the current technical security of an online critical illness crowdfunding platform, the more perfect the public will believe the interactive design of the online platform perfec to be. The more open and transparent the

fundraising process and the destination of donations, the more the public's trust in the platform will increase. This increase will clarify the donation attitude that donations through critical illness crowdfunding network platforms can help the target audience, thereby enhancing the public's willingness to donate, which in turn will lead to donations. This study focuses on interactivity and discovers the relationships between interactivity and platform trust, which further influence donation attitude, which in turn affects donation behavior. This study broadens our understanding of the influence of trust on donation behavior and lays a theoretical basis for subsequent related research.

**Innovatively introducing empathy and social distance as variables for exploring the influence of individual characteristics on donation behavior.** (1) This study finds that empathic concern positively affects donation attitude, which in turn affects donation behavior. That is, when donors think that help-seekers are similar to themselves, they are more likely to have the emotion of sadness and are more inclined to donate. The collectivist culture of China emphasizes the interdependence between individuals, which encourages individuals to increase their willingness to help others based on their sympathetic responses to others [100]. (2) This study also confirms that social distance positively affects donation willingness, which in turn affects donation behavior. From a sociological perspective, social distance reflects the characteristics of the psychological distance between oneself and others or groups, which in turn describes the closeness of the relationship between them. When an individual thinks that there is a small social distance, he or she is more inclined to make donations [75]. The social distance variable with Chinese social and cultural characteristics explores the influence of this personal factor on donation attitude and behavior, and it enriches existing theoretical research models and lays a foundation for subsequent research.

## Acknowledgments

We would like to thank all the participants who completed the questionnaires.

## Author Contributions

**Conceptualization:** Lu Chen.

**Data curation:** Fan Luo.

**Investigation:** Wanshi He.

**Methodology:** Lu Chen, Fan Luo.

**Project administration:** Lu Chen.

**Software:** Fan Luo.

**Visualization:** Wanshi He, Heng Zhao, Liru Pan.

**Writing – original draft:** Wanshi He, Heng Zhao.

**Writing – review & editing:** Liru Pan.

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
