## [Decision Letter · Decision Letter 0]

18 Mar 2021

PONE-D-21-03866

A study on the influencing factors of the public's willingness to donate funds for critical illness crowdfunding projects on network platforms

PLOS ONE

Dear Dr. Chen,

Thank you for submitting your manuscript to PLOS ONE. After careful consideration, we feel that it has merit but does not fully meet PLOS ONE’s publication criteria as it currently stands. Therefore, we invite you to submit a revised version of the manuscript that addresses the points raised during the review process.

The manuscript needs a MAJOR REVISION in order to be evaluated for a future publication. Please, have a look to the suggestions provided by the reviewers. An important improvement in the theoretical aspect of the paper is needed.

We look forward to receiving your revised manuscript.

Kind regards,

Barbara Guidi

Academic Editor

PLOS ONE

Journal Requirements:

5. We note that Figure 3 in your submission contain map images which may be copyrighted. All PLOS content is published under the Creative Commons Attribution License (CC BY 4.0), which means that the manuscript, images, and Supporting Information files will be freely available online, and any third party is permitted to access, download, copy, distribute, and use these materials in any way, even commercially, with proper attribution. For these reasons, we cannot publish previously copyrighted maps or satellite images created using proprietary data, such as Google software (Google Maps, Street View, and Earth). For more information, see our copyright guidelines: http://journals.plos.org/plosone/s/licenses-and-copyright.

5.1.    You may seek permission from the original copyright holder of Figure 3 to publish the content specifically under the CC BY 4.0 license. 

5.2.    If you are unable to obtain permission from the original copyright holder to publish these figures under the CC BY 4.0 license or if the copyright holder’s requirements are incompatible with the CC BY 4.0 license, please either i) remove the figure or ii) supply a replacement figure that complies with the CC BY 4.0 license. Please check copyright information on all replacement figures and update the figure caption with source information. If applicable, please specify in the figure caption text when a figure is similar but not identical to the original image and is therefore for illustrative purposes only.

Reviewers' comments:

Reviewer's Responses to Questions

**Comments to the Author**

1. Is the manuscript technically sound, and do the data support the conclusions?

Reviewer #1: Yes

Reviewer #2: Yes

Reviewer #3: No

2. Has the statistical analysis been performed appropriately and rigorously? 

Reviewer #1: Yes

Reviewer #2: Yes

Reviewer #3: No

3. Have the authors made all data underlying the findings in their manuscript fully available?

Reviewer #1: No

Reviewer #2: Yes

Reviewer #3: Yes

4. Is the manuscript presented in an intelligible fashion and written in standard English?

Reviewer #1: Yes

Reviewer #2: Yes

Reviewer #3: No

5. Review Comments to the Author

Reviewer #1: Manuscript PONE-D-21-03866 is a study of the influencing factors of the public willingness to donate funds to critical

illness via crowd-funding platforms.

They use a well established framework of structural equation modeling (SEM) evalute path coefficient for 14 well grounded hypotheses.

1) The manuscript would benefit from exploratory data analysis (visual). For example, it would be illustrative to see the distributions of potential variables to confirm their "normality". Of course in this particular case the variables are categorigal and SEM can be used only approximately, which should mentioned in the analysis. Another example would be visualing correlation between varios external variables (gender, age and income) and potential variables.

2) It would be great to evaluating the effect of gender, age and income level on the path coefficients. What if there is significant divergence there? For example if between two groups one has positive effect and the other negative, the overall result could be close to zero, as in H1.

Overall after a minor revision the manuscript is suitable for publication.

Reviewer #2: This is obviously an interesting article. However, there are few issues that need to looked at. Find the following:

1. For example, the article mentions both "Internet +" donations and "Internet donations". How different are these two concepts?

2. Besides, it would have been helpful especially for readers who may not be conversant with this model of donation to know exactly what "Internet +" is. Is it generally known or it's restricted to only the study area?

3. It may be helpful to provide distinctions among the 12 latent variables that were considered. How different are they from each other?

4. It was also not to clear the target of the recommendations. The suggestion is for the recommendations to be targeted at particular person/institution etc.

Reviewer #3: The authors are quite ambitious to use 45 variables with 12 latent factors in a SEM (n = 710) to explain the behaviour related to crowdfunding projects on network platform in China. However, the proposed model failed to fulfil the minimum criteria for adequate model fit, with merely GFI = 0.788, CFI = 0.835 and X2/df = 3.909 (p. 35-36), i.e. the entire manuscript needs to be re-written, with revised research questions, hypothesis, theoretical framework, and data analysis.

More importantly, the theoretical framework is very weak, the authors only list out various theories and perspectives (p. 12-23) but without showing any linkage with reference to the existing literature.

The conclusions and recommendations, such as urging for more governmental supervision, by using legal framework and blockchain technology for further government monitoring etc., did not based on the results/empirical findings.

Hence, I do not think the current form of manuscript is suitable for publication. There are major flaws in the research design and data analysis methods. There is also lack of theoretical contribution.

6. PLOS authors have the option to publish the peer review history of their article (what does this mean?). If published, this will include your full peer review and any attached files.

Reviewer #1: No

Reviewer #2: No

Reviewer #3: **Yes: **Sai-fu Fung

---

## [Author Response · Author response to Decision Letter 0]

25 May 2021

Response to Reviewers

Dear Editor and Reviewers,

Re: Manuscript PONE-D-21-03866, A study on the influencing factors of the public's willingness to donate funds for critical illness crowdfunding projects on network platforms.

Please find attached a revised version of our manuscript “PONE-D-21-03866”, which we would like to resubmit for publication as a Original Research in PLoS ONE.

Your comments were highly insightful and enabled us to greatly improve the quality of our manuscript. In the following pages are our point-by-point responses to each of the comments of the reviewers as well as your own comments.

Revisions in the text are shown using red highlight for additions. 

According to the recommendation of reviewer 1, 2 and 3, we will reply to the opinions of the three reviewers below, including what specific improvements we have made to the manuscript according to the opinions of the reviewers.

Similarly, after revising according to the reviewers' opinions, other parts of the full text are modified accordingly, such as the abstract, literature review and theoretical part, and a series of tables and figures are also included.

If there are still areas for improvement, we are willing to seriously revise them as soon as possible.

What needs to be added is that there is a commen in the review comments that our English expression is not standardized, but it needs to be explained. When we submitted the first manuscript, we already chose the editing agency suggested by PLoS and the manuscript was edited by one or more of the highly qualified native English speaking editors at AJE. And attach a editing certificate. We don't know which part of the problem led to such comment.

Once again, we sincerely thank you for your enthusiastic work, your careful evaluation and patient guidance. We learned a lot. It was a very pleasant experience.

We hope that the revision of the manuscript and our reply will be enough to make our manuscript suitable for publication in PLoS ONE.

We look forward to hearing from you at your earliest convenience.

Yours sincerely,

Lu Chen

CATALOGUE

Response to Reviewer #1 3

Response to Reviewer #2 6

Response to Reviewer #3 9

Response to Journal Requirements 12

Reviewer #1: 

Manuscript PONE-D-21-03866 is a study of the influencing factors of the public willingness to donate funds to critical illness via crowd-funding platforms.

They use a well established framework of structural equation modeling (SEM) evalute path coefficient for 14 well grounded hypotheses.

1)The manuscript would benefit from exploratory data analysis (visual). For example, it would be illustrative to see the distributions of potential variables to confirm their "normality". Of course in this particular case the variables are categorigal and SEM can be used only approximately, which should mentioned in the analysis.Another example would be visualing correlation between varios external variables (gender, age and income) and potential variables.

2) It would be great to evaluating the effect of gender, age and income level on the path coefficients. What if there is significant divergence there? For example if between two groups one has positive effect and the other negative, the overall result could be close to zero, as in H1.

Overall after a minor revision the manuscript is suitable for publication.

Our replies:

Thank you very much for your careful review and constructive suggestions with regard to our manuscript.

Thoses comments are very helpful for us to revise and improve our paper.

There are two main aspects of reviewer 1's comments, which are to be answered one by one.

1.For the first suggestion, we have observed the distributions of potential variables and determined that they are all approximately "normality". Of course, in this special case, variables are classified, SEM can only be used approximately, We also emphasized this sentence in the revised manuscript（The details are shown from line 503~505）

A— Donation attitude；C— Empathic Concern；E— Perceived usefulness；G— Platform trust；

J— Interactivity；k— Donation willingness；L— Social distance；M— Donation behavior

2.For the second question, we have evaluated the effects of gender, age and income level on the path coefficients. Through grouping observation, We find that there is no significant difference in the overall model path through grouping observation, and some paths, such as men, are negative, but not significant. The above reasons are due to the differences of samples, that is, there are differences in cognition and behavior among different groups.

We hope that the corrections will meet with your concerns. 

Special thanks to you for your good comments. 

Reviewer #2: 

This is obviously an interesting article. However, there are few issues that need to looked at.

Find the following:

1. For example, the article mentions both "Internet +" donations and "Internet donations". How different are these two concepts?

2. Besides, it would have been helpful especially for readers who may not be conversant with this model of donation to know exactly what "Internet +" is. Is it generally known or it's restricted to only the study area?

3. It may be helpful to provide distinctions among the 12 latent variables that were considered. How different are they from each other?

4. It was also not to clear the target of the recommendations. The suggestion is for the recommendations to be targeted at particular person/institution etc.

Our replies:

Thank you very much for your careful review and constructive suggestions with regard to our manuscript. Thoses comments are very helpful for us to revise and improve our paper.

There are three main aspects of reviewer 2's comments, which are to be answered one by one.

1.The first and second questions you mentioned, especially about the articles both "Internet +" donations and "Internet donations", we also found this kind of problems when we checked them. There is no obvious difference between the two, but it does cause ambiguity to readers. In the revised manuscript, there has been a unified way of expression, namely "Internet donations". Thank you for your advice.

2.There are mainly eight variables in the revised and improved model. No matter the previous 12 variables or the current 8 variables, they are mainly based on the two classic models in the field of consumer behavior research——TAM and TPB.

The reason is that TAM can explore the trade-off process when the public uses the platform as a system function. As a general model, TAM only provides two cognitive concepts that affect individuals' willingness to accept technology, namely perceived usefulness and perceived ease of use. It does not explain or limit the external variables that affect the usefulness and ease of use in specific application situations, while TPB combines various variables, including the public's attitude towards behavior and the influence of others.

Therefore, in the TAM model, in order to better explain the public's willingness to donate and decision-making behavior under the specific circumstances of the critical illness crowdfunding platform, the public's trust in the platform is taken as an important premise of perceived usefulness, which is based on the design of the interactive mechanism provided by the platform. 

TAM model itself includes four variables: perceived usefulness, donation attitude, donation willingness and donation behavior.

Personal empathy is also added as a variable of personal factors to measure the public's willingness to donate to the crowdfunding projects for major diseases on the Internet platform.

In addition, in the part of TPB model, the TPB holds that the behavior of an individual is not always controlled by his or her own will. Rather, the individual's behavior is affected by internal and external factors. Attitudes and standardized responses to performed situations will change the individual's behavior, so we added the variable of social distance.

Therefore, there are eight variables.

（The details are shown from line 108~372）

3.For the fourth question, thank you very much for your suggestions, including your suggestion that the suggestion is for the recommendations to be targeted at particular person / institution, etc

We have readjusted this part according to your suggestions, which can be divided into suggestions for the government, platforms and help-seekers.

（The details are shown from line 611~708）

We hope that the corrections will meet with your concerns. 

Special thanks to you for your good comments. 

Reviewer #3: 

The authors are quite ambitious to use 45 variables with 12 latent factors in a SEM (n = 710) to explain the behaviour related to crowdfunding projects on network platform in China. However, the proposed model failed to fulfil the minimum criteria for adequate model fit, with merely GFI = 0.788, CFI = 0.835 and X2/df = 3.909 (p. 35-36), i.e. the entire manuscript needs to be re-written, with revised research questions,

hypothesis, theoretical framework, and data analysis.

More importantly, the theoretical framework is very weak, the authors only list out various theories and perspectives (p. 12-23) but without showing any linkage with reference to the existing literature.

The conclusions and recommendations, such as urging for more governmental supervision, by using legal framework and blockchain technology for further government monitoring etc., did not based on the results/empirical findings.

Hence, I do not think the current form of manuscript is suitable for publication. There are major flaws in the research design and data analysis methods. There is also lack of theoretical contribution.

Our replies:

Thank you very much for your careful review and constructive suggestions with regard to our manuscript. Thoses comments are very helpful for us to revise and improve our paper. 

There are three main aspects of reviewer 3's comments, which are to be answered one by one.

1.The first question is that the proposed model does not meet the fitting standard. 

With a little explanation of this situation, in the first draft, there were 12 variables considered at that time, and the whole model was more complex. Therefore, the requirements for individual index data were lowered, only close to 0.9, but not more than 0.9. It was really worth reflecting that they were reported as model fitting results.

To solve this problem, in this draft, our revision ideas and specific measures are as follows:

Reorganize variables and models. In the original TAM model, a few variables, such as perceived ease of use, which are not suitable for current situation discussion, are eliminated. In traditional TAM, the higher the ease of use of a critical illness crowdfunding platform perceived by the public is, the higher the usefulness and effectiveness of the platform perceived by the public. However, the ease of use of the Internet platform has been very high after years of development, So the variable of the ease of use will not be considered in this study. At the same time, considering that both subjective norm and social distance measure the influence of the surrounding groups on the donors, we choose the variable of social distance in the new draft based on the theoretical basis and practical observation. To sum up, there are 8 variables remaining in the model and 8 hypotheses proposed. The model fits well and all the path hypotheses are valid.

（The details are shown from line 512~517）

2.The second question is about the insufficient theory.

In the process of revising the manuscript, we still rescreened the relevant literature, but the relevant literature is still not rich.

(1) There are few empirical studies on this topic in China, and there are few high-level literatures;

(2) Foreign related literature is not much, some donation articles mainly focus on organ donation;

(3) Although there are not many high-level literatures for reference, it is still a social phenomenon and hot issue of great research significance. Therefore, we start from the perspective of consumer behavior and choose the classic TAM and TPB models. The variables commonly used in these two models are relatively fixed, and the literatures on these two models are very common, Therefore, there is not too much theoretical elaboration in the article, and the article introduces the theory and literature basis of path hypothesis in detail. As far as the current structure is concerned, both the literature review and the path hypothesis are fairly long. If the reviewers feel that it is still necessary to explain the TAM and TPB variables in detail, we will complete it as soon as possible.

3.The third question is about policy recommendations. In this draft, we made the following adjustments:

(1) Readjust the structure and level of the proposal part. It can be divided into for government, platforms and help-seekers;

(2) As for the government's proposal you mentioned, which has no basis, we have also made an explanation and response. The proposal to strengthen government supervision, such as improving legislation, is based on the review of the current laws and government policy reports. The reason why we didn't launch the discussion is that the theme of the discussion is the influencing factors and paths of donation behavior. Of course, if we all agree that we should consolidate here, we will continue to work hard.

We hope that the corrections will meet with your concerns.

Special thanks to you for your good comments.

Response to Journal Requirements

Dear Editor,

I'm very glad to receive your email. It's very nice of you to proofread our manuscript. The following part is about some of the necessity of Journal Requirements.

According to your suggestions, we have responded one by one and adjusted the corresponding parts in the revised manuscript, and our replies are marked in yellow.

Thank you again for your hard work.

If there is anything wrong, please let me know. 

Thank you.

Thank you for your suggestion. The revised manuscript has been received according to the style requirements of PLoS One to arrange the chapters and layout. If there is anything wrong, we will improve it according to the requirements of the journal.

2.Please include additional information regarding the survey or questionnaire used in the study and ensure that you have provided sufficient details that others could replicate the analyses. For instance, if you developed a questionnaire as part of this study and it is not under a copyright more restrictive than CC-BY, please include a copy, in both the original language and English, as Supporting Information.

Thank you for your reminding. We have provided enough details about the survey or questionnaire used in the study in Table 1, and others can copy our questionnaire and analyze it.

3. We note that you have indicated that data from this study are available upon request. PLOS only allows data to be available upon request if there are legal or ethical restrictions on sharing data publicly. For more information on unacceptable data access restrictions, please see 

http://journals.plos.org/plosone/s/data-availability#loc-unacceptable-data-access-restrictions.

a) If there are ethical or legal restrictions on sharing a de-identified data set, please explain them in detail

(e.g., data contain potentially sensitive information, data are owned by a third-party organization, etc.) and

who has imposed them (e.g., an ethics committee). Please also provide contact information for a data access

committee, ethics committee, or other institutional body to which data requests may be sent.

b) If there are no restrictions, please upload the minimal anonymized data set necessary to replicate your

study findings as either Supporting Information files or to a stable, public repository and provide us with the

relevant URLs, DOIs, or accession numbers. For a list of acceptable repositories, please see

http://journals.plos.org/plosone/s/data-availability#loc-recommended-repositories.

We have uploaded the minimum anonymous data set, which includes the values behind the reported average, standard deviation and other measures, as well as the values used to build the graph(in Table 1 ~ Table 5）

4. PLOS requires an ORCID iD for the corresponding author in Editorial Manager on papers submitted after

December 6th, 2016. Please ensure that you have an ORCID iD and that it is validated in Editorial Manager. To

do this, go to ‘Update my Information’ (in the upper left-hand corner of the main menu), and click on the

Fetch/Validate link next to the ORCID field. This will take you to the ORCID site and allow you to create a new

iD or authenticate a pre-existing iD in Editorial Manager. Please see the following video for instructions on

linking an ORCID iD to your Editorial Manager account: 

https://www.youtube.com/watch?v=_xcclfuvtxQ

Thank you for your reminding. Our three corresponding authors have updated their ORCID ID, which are:

Fan Luo https://orcid.org/0000-0002-8487-0200

Wanshi He https://orcid.org/0000-0001-8109-6566

Heng Zhao https://orcid.org/0000-0002-7344-4329

5. We note that Figure 3 in your submission contain map images which may be copyrighted. All PLOS

content is published under the Creative Commons Attribution License (CC BY 4.0), which means that the

manuscript, images, and Supporting Information files will be freely available online, and any third party is

permitted to access, download, copy, distribute, and use these materials in any way, even commercially, with

proper attribution. For these reasons, we cannot publish previously copyrighted maps or satellite images

created using proprietary data, such as Google software (Google Maps, Street View, and Earth). For more

information, see our copyright guidelines: 

http://journals.plos.org/plosone/s/licenses-and-copyright.

Thank you for your reminding. Considering that the figure is not important to the full text, so we deleted the figure in the revised manuscript.

---

## [Decision Letter · Decision Letter 1]

14 Jun 2021

PONE-D-21-03866R1

A study on the influencing factors of the public's willingness to donate funds for critical illness crowdfunding projects on network platforms

PLOS ONE

Dear Dr. Chen,

Thank you for submitting your manuscript to PLOS ONE. After careful consideration, we feel that it has merit but does not fully meet PLOS ONE’s publication criteria as it currently stands. Therefore, we invite you to submit a revised version of the manuscript that addresses the points raised during the review process.

The paper needs a MAJOR REVISION. Authors should revised the manuscript in order to address all the suggestions given by the reviewers.

We look forward to receiving your revised manuscript.

Kind regards,

Barbara Guidi

Academic Editor

PLOS ONE

Reviewers' comments:

Reviewer's Responses to Questions

**Comments to the Author**

1. If the authors have adequately addressed your comments raised in a previous round of review and you feel that this manuscript is now acceptable for publication, you may indicate that here to bypass the “Comments to the Author” section, enter your conflict of interest statement in the “Confidential to Editor” section, and submit your "Accept" recommendation.

Reviewer #1: (No Response)

Reviewer #3: (No Response)

2. Is the manuscript technically sound, and do the data support the conclusions?

Reviewer #1: No

Reviewer #3: Partly

3. Has the statistical analysis been performed appropriately and rigorously? 

Reviewer #1: No

Reviewer #3: Yes

4. Have the authors made all data underlying the findings in their manuscript fully available?

Reviewer #1: Yes

Reviewer #3: Yes

5. Is the manuscript presented in an intelligible fashion and written in standard English?

Reviewer #1: Yes

Reviewer #3: Yes

6. Review Comments to the Author

Reviewer #1: Manuscript PONE-D-21-03866 is a study of the influencing factors of the public willingness to donate funds to critical

illness via crowd-funding platforms.

They use a well established framework of structural equation modeling (SEM) evaluate path coefficient for 14 well grounded hypotheses.

The authors did not adequately answer questions 1 and 2 raised previously.

1) On 137 the authors plot the distribution and claim that they observe that they are normal. It is customary to use a KS or D’Agostino and Pearson’s test to test whether a distribution is normal. It clear from the figures most of the distributions are not normal, which invalidates the chosen approach. The second part regarding the visualization of interdependence of available features has been ignored.

2) The response to the second question is also verbal. No figures or metrics were provided.

Quantitative estimates of 1) and 2) should appear in the main text, if they were made.

I conclude that the manuscript has not been brought to the requested level of clarity. A major revision is necessary.

Reviewer #3: Thanks for submitting the revised manuscript to address my concerns. However, the authors still failed to fully address my comments in point number 1, 2 and 3:

1) With regarded to the first point, the SEM result is now fulfilled the criteria for model fit, however, please provide more details about the estimator used (Li, 2016)? Did the model with any error terms correlated (Hermida, 2015)?

2) The discussion of the theoretical framework is still very thin, in addition to the TAM and TPB, the authors still need to provide theoretical justifications for including other variable in the model, such as empathy, trust, planning behavior and social distance. Developing a ‘model’ without much theoretical foundations are risky and dangerous, see Burghardt and Bodansky (2021) and Borsboom, van der Maas, Dalege, Kievit, and Haig (2021), and (DeYoung & Krueger, 2020).

3) As the authors mentioned, the proposal to strength government supervision was merely ‘based on the review of the current laws and government policy reports’. Any recommendations that is not based on the study’s findings should be removed to avoid any confusions to the reader.

References

Borsboom, D., van der Maas, H. L. J., Dalege, J., Kievit, R. A., & Haig, B. D. (2021). Theory Construction Methodology: A Practical Framework for Building Theories in Psychology. Perspectives on Psychological Science, 0(0), 1745691620969647. doi:10.1177/1745691620969647

Burghardt, J., & Bodansky, A. N. (2021). Why Psychology Needs to Stop Striving for Novelty and How to Move Towards Theory-Driven Research. Frontiers in Psychology, 12(67). doi:10.3389/fpsyg.2021.609802

DeYoung, C. G., & Krueger, R. F. (2020). To Wish Impossible Things: On the Ontological Status of Latent Variables and the Prospects for Theory in Psychology. Psychological Inquiry, 31(4), 289-296. doi:10.1080/1047840X.2020.1853462

Hermida, R. (2015). The problem of allowing correlated errors in structural equation modeling: concerns and considerations. Computational Methods in Social Sciences (CMSS), 3(1), 05-17. Retrieved from https://EconPapers.repec.org/RePEc:ntu:ntcmss:vol3-iss1-15-005

Li, C. H. (2016). Confirmatory factor analysis with ordinal data: Comparing robust maximum likelihood and diagonally weighted least squares. Behavior Research Methods, 48(3), 936-949. doi:10.3758/s13428-015-0619-7

7. PLOS authors have the option to publish the peer review history of their article (what does this mean?). If published, this will include your full peer review and any attached files.

Reviewer #1: No

Reviewer #3: **Yes: **Sai-fu Fung

---

## [Author Response · Author response to Decision Letter 1]

24 Aug 2021

Response to Reviewers

Dear Editor and Reviewers,

Re: Manuscript PONE-D-21-03866R1, A study on the influencing factors of the public's willingness to donate funds for critical illness crowdfunding projects on network platforms.

Please find attached a revised version of our manuscript “PONE-D-21-03866R1”, which we would like to resubmit for publication as a Original Research in PLoS ONE.

Your comments were highly insightful and enabled us to greatly improve the quality of our manuscript. In the following pages are our point-by-point responses to each of the comments of the reviewers as well as your own comments.

Revisions in the text are shown using red highlight for additions. 

First of all, thank you very much for the comments of the editor and reviewers. We have also made significant revisions to the paper. Now we will give a brief description of the work we have done, and then answer the comments of reviewers one by one in the second part.

1. In this part, nearly 20 new literatures have been added. Of course, necessary adjustments have been made to the logical expression. In the revised version with a trajectory, it is marked in red.

 2. In the demonstration part, this draft is clearly divided into the EFA part and the CFA part, and relevant tables and figures are supplemented.

(1) Some necessary analyses were added to the Validity Test section. Skewness and kurtosis can effectively describe the normal distribution of data for each observed variable. The values of Skewness and kurtosis are shown in Table 3 within ±1. The absolute value of each skewness is less than 3, and the absolute value of each kurtosis is less than 10. All variables and averages are in accordance with the standard, which indicates the data shows a normal distribution.

(2) Secondly, Factor analysis in "Table 4. Rotated factor Analysis (factor analysis)". As can be seen from Table 4, the exploratory factor loads of 31 indicators were obtained through the maximum variance rotation analysis. These quantities are all greater than 0.5, satisfying the model requirements and allowing the extraction of 8 factors to explain the structure of the variables.

(3) The factor analysis of Confirmatory was improved. See Figure 3 starting at line 610 for details.

According to the suggestions, we have drawn detailed Path Coefficients of the Hypothetical Model. See Fig.4 in line 639.

3. Delete all recommendations that are not based on the study's findings, and make necessary streamlining of the summary, introduction and conclusion according to the adjustment of the main body.

The following are the responses to the suggestions of reviewers respectively:

Reviewer #1: 

The authors did not adequately answer questions 1 and 2 raised previously.

1) On 137 the authors plot the distribution and claim that they observe that they are normal. It is customary to use a KS or D’Agostino and Pearson’s test to test whether a distribution is normal. It clear from the figures most of the distributions are not normal, which invalidates the chosen approach. The second part regarding the visualization of interdependence of available features has been ignored.

2) The response to the second question is also verbal. No figures or metrics were provided.

Quantitative estimates of 1) and 2) should appear in the main text, if they were made.

I conclude that the manuscript has not been brought to the requested level of clarity. A major revision is necessary.

Response:

Thanks for the reviewer's comments. On behalf of the team, I would like to apologize for our negligence in the last draft. As for the two problems you mentioned, we have carefully discussed and revised them in this draft.

1. The first question has been reflected in the text of this draft, for details, see 2 (1) in the answer section.

Some necessary analyses were added to the Validity Test section. Skewness and kurtosis can effectively describe the normal distribution of data for each observed variable. The values of Skewness and kurtosis are shown in Table 3 within ±1. The absolute value of each skewness is less than 3, and the absolute value of each kurtosis is less than 10. All variables and averages are in accordance with the standard, which indicates the data shows a normal distribution.

2. The second question is whether gender, age and income have a significant effect on the path coefficient.

The process of this analysis is not reflected in the current manuscript, but we have verified it. If necessary, we are also very happy to supplement the manuscript as soon as possible. The idea is as follows: 

Divergence: We divided gender into men and women, aged under 30 years old and over 30 years old, and income into under 5000 yuan and over 5000 yuan, and then looked at the path coefficient respectively, but there was no significant difference in general. The age and income division takes into account the actual situation of the respondents, including the old Chinese saying "one should stand firm at the age of 30" and the average per capita income level of Chinese cities.

One path coefficient of gender is -0.071, which can be explained by the fact that women are more likely to have empathy than men when they want to donate, which is also consistent with some literature studies [1]. Sisco and Weber analyzed the influencing factors of online donation behavior on the GoFundMe platform, and their survey found that women are more likely to resonate with donation information [2].

The income path coefficient of -0.024 can be explained as well-off donors are more likely to make donations, which is also consistent with some literature studies [3-5].

Thanks again for the reviewer's comments, which are of great help to us. We also hope that our revision this time is consistent with your suggestions.

Gender

Fig 1.(Detailed figures in the response to reviews file)

Age

Fig 2.(Detailed figures in the response to reviews file)

Income

Fig 3.(Detailed figures in the response to reviews file)

Reference:

[1]Mesch,D.J., Brown,M.S, Moore,Z.l,& Hayat,A.D.(2011). Gender differences in charitable giving. International Journal of Nonprofit and Voluntary Sector Marketing,16(4),342-355. doi:10.1002/nvsm.432 

[2]Sisco MR, Weber EU. Examining charitable giving in real-world online donations. Nat Commun. 2019;10: 3968.

[3] Schwienbacher A, Larralde B. Crowdfunding of small entrepreneurial ventures. In: Cumming D, editor. The Oxford handbook of entrepreneurial finance. Oxford, UK: Oxford University Press; 2012. pp. 369-391.

[4] Lin Z, Xiao Q, Zhou Z. An empirical study on the relationship between ethical predispositions and charitable behavior: Based on the moderating effect of moral identity. Foreign Econ Manag. 2014;36: 16-31.

[5] Hui J, Gerber E, Greenberg M. Easy money? The demands of crowdfunding work. Segal technical report. 1st ed. Evanston, IL: Northwestern University; 2012.

Reviewer #3: 

 However, the authors still failed to fully address my comments in point number 1, 2 and 3:

1) With regarded to the first point, the SEM result is now fulfilled the criteria for model fit, however, please provide more details about the estimator used (Li, 2016)? Did the model with any error terms correlated (Hermida, 2015)?

2) The discussion of the theoretical framework is still very thin, in addition to the TAM and TPB, the authors still need to provide theoretical justifications for including other variable in the model, such as empathy, trust, planning behavior and social distance. Developing a ‘model’ without much theoretical foundations are risky and dangerous, see Burghardt and Bodansky (2021) and Borsboom, van der Maas, Dalege, Kievit, and Haig (2021), and (DeYoung & Krueger, 2020).

3) As the authors mentioned, the proposal to strength government supervision was merely ‘based on the review of the current laws and government policy reports’. Any recommendations that is not based on the study’s findings should be removed to avoid any confusions to the reader.

Response:

Thank you very much for the reviewer's patient guidance. As for the three suggestions you mentioned, I would like to review our revision work on behalf of the team.

1.For the first question, provide more details about the estimator used. 

We have completed drawing detailed Path Coefficients of the Hypothetical Model. See Fig.4 in line 639. Thank you.

2.Second question, thank you for your reminding.We strongly agree with your suggestions. On this issue, we are also very discreet. On the one hand, on the basis of the literature part before, we have collected the latest literature and omissions to enhance the theoretical basis, nearly more than 20 new references. See the red section for specific new content. On the other hand, we have carried on the exploratory factor analysis, through the principal component analysis ie. and a series of measures to ensure that the model construction is “mathematically” correct .For the third problem, this is a good suggestion. In this draft, all recommendations that are not based on the study's findings are deleted, and after the adjustment in the main body, the summary, introduction and conclusion are all simplified accordingly.

If there are still areas for improvement, we are willing to seriously revise them as soon as possible.

Once again, we sincerely thank you for your enthusiastic work, your careful evaluation and patient guidance. We learned a lot. It was a very pleasant experience.

We hope that the revision of the manuscript and our reply will be enough to make our manuscript suitable for publication in PLoS ONE.

We look forward to hearing from you at your earliest convenience.

Yours sincerely,

Lu Chen

---

## [Decision Letter · Decision Letter 2]

20 Sep 2021

PONE-D-21-03866R2A study on the influencing factors of the public's willingness to donate funds for critical illness crowdfunding projects on network platformsPLOS ONE

Dear Dr. Chen,

Thank you for submitting your manuscript to PLOS ONE. After careful consideration, we feel that it has merit but does not fully meet PLOS ONE’s publication criteria as it currently stands. Therefore, we invite you to submit a revised version of the manuscript that addresses the points raised during the review process.

The paper needs a MINOR REVISION. Please, address all the requests in order to improve the readability of the paper, and to improve the evaluation process.

We look forward to receiving your revised manuscript.

Kind regards,

Barbara Guidi

Academic Editor

PLOS ONE

Journal Requirements:

Reviewers' comments:

Reviewer's Responses to Questions

**Comments to the Author**

1. If the authors have adequately addressed your comments raised in a previous round of review and you feel that this manuscript is now acceptable for publication, you may indicate that here to bypass the “Comments to the Author” section, enter your conflict of interest statement in the “Confidential to Editor” section, and submit your "Accept" recommendation.

Reviewer #1: All comments have been addressed

Reviewer #3: (No Response)

2. Is the manuscript technically sound, and do the data support the conclusions?

Reviewer #1: No

Reviewer #3: Partly

3. Has the statistical analysis been performed appropriately and rigorously? 

Reviewer #1: No

Reviewer #3: No

4. Have the authors made all data underlying the findings in their manuscript fully available?

Reviewer #1: Yes

Reviewer #3: Yes

5. Is the manuscript presented in an intelligible fashion and written in standard English?

Reviewer #1: Yes

Reviewer #3: Yes

6. Review Comments to the Author

Reviewer #1: The normality assumption had not been addressed properly.

I have already suggested to D'Agostino test for normality that yields a p-value that had been ignored.

"Skewness and kurtosis can effectively describe the normal distribution of data for each observed variable". - a very odd statement. Which is not followed by any statistical claim.

Reviewer #3: Thanks for submitting the revised manuscript to further address my concerns. But there are still two outstanding issues need to be addressed and also related to my previous comments:

1) It seems the authors are using the CFA goodness-of-fit results (Figure 3, p. 29 and Table 6, p. 31) to justify the SEM (Figure 4, p. 32) also fulfil the criteria of adequate model fit. The authors said that ‘As shown in Table 6, the same indices can be used to evaluate of both the CFA and SEM’ (p. 32, lines 646-647). I am afraid that there is a serious methodological flaw here. The proposed SEM model and the sequences results (Table 7) cannot be accepted, unless the authors can show the SEM results’ fit indices of Figure 4. The authors should also submit the anonymous raw data and syntax used to enable to readers to replicate the results.

2) For the discussion section, the authors should further discuss the major findings according to the existing theory and the literature. The entire section (p. 35-37) without any literature to support the discussion is a bit lack of scientific rigor.

7. PLOS authors have the option to publish the peer review history of their article (what does this mean?). If published, this will include your full peer review and any attached files.

Reviewer #1: No

Reviewer #3: **Yes: **Sai-fu Fung

---

## [Author Response · Author response to Decision Letter 2]

17 Dec 2021

Response to Reviewers

Dear Reviewers,

Re: Manuscript PONE-D-21-03866R2, A study on the influencing factors of the public's willingness to donate funds for critical illness crowdfunding projects on network platforms.

Please find attached a revised version of our manuscript “PONE-D-21-03866R2”, which we would like to resubmit for publication as a Original Research in PLoS ONE.

Your comments were highly insightful and enabled us to greatly improve the quality of our manuscript. In the following pages are our point-by-point responses to each of the comments of the reviewers as well as your own comments.

Revisions in the text are shown using red highlight and track changes for additions. 

First of all, thank you very much for the comments of the editor and reviewers. We have also made significant revisions to the paper. Now we will answer the comments of reviewers one by one in the second part.

The following are the responses to the suggestions of reviewers respectively:

Reviewer #1: 

The normality assumption had not been addressed properly.

I have already suggested to D'Agostino test for normality that yields a p-value that had been ignored.

"Skewness and kurtosis can effectively describe the normal distribution of data for each observed variable". - a very odd statement. Which is not followed by any statistical claim.

Response:

Thanks for the reviewer's comments. 

As for your suggestions, we have carefully considered them in the last manuscript revision, including the D 'Agostino test of normal distribution you mentioned. In our last manuscript, we used "Skewness and Kurtosis" to check the normal distribution for the following two reasons. 

First, we checked the journal Jiang W, Huang Z, Peng Y, Fang Y, Cao Y published in PLoS ONE (2020) Factors affecting Prefabricated construction Promotion in China: A structural equation modeling. PLoS ONE 15(1): E0227787.

In his paper, the method of "Skewness and Kurtosis "was also adopted and specific references were given (No. 97 reference in this manuscript).

At the same time, in order to further confirm whether this method is a good method to verify the normal distribution, we also refer to literatures with up to 315 citations: Hopkins K D, Weeks D L. Tests for Normality and Measures of Skewness and Kurtosis:Their place in research reporting[J]. Educational and Psychological Measurement, 1990, 50(4): 717-729. (No. 98 reference in this manuscript) 

In this paper Page 721 mentioned ” In addition to an omnibus test of normality,separate measures and tests of skewness and kurtosis have descriptive and inferential value. (Some statisticians recommend that separate inferential tests for skewness and kurtosis be used only after an omnibus test has rejected the normality assumption, but Monte Carlo studies have demonstrated that with certain types of distributions the separate tests(eqs. 5 and 8 )have greater power than even the Shapiro-wilk test(Shapiro et al., 1968).)”

Based on the above reasons, we’ve completed the normal distribution test follow the experience of papers published in the same journal(PLoS ONE) in the manuscript.

Reviewer #3: 

Thanks for submitting the revised manuscript to further address my concerns. But there are still two outstanding issues need to be addressed and also related to my previous comments:

1) It seems the authors are using the CFA goodness-of-fit results (Figure 3, p. 29 and Table 6, p. 31) to justify the SEM (Figure 4, p. 32) also fulfil the criteria of adequate model fit. The authors said that ‘As shown in Table 6, the same indices can be used to evaluate of both the CFA and SEM’ (p. 32, lines 646-647). I am afraid that there is a serious methodological flaw here. The proposed SEM model and the sequences results (Table 7) cannot be accepted, unless the authors can show the SEM results’ fit indices of Figure 4. The authors should also submit the anonymous raw data and syntax used to enable to readers to replicate the results.

2) For the discussion section, the authors should further discuss the major findings according to the existing theory and the literature. The entire section (p. 35-37) without any literature to support the discussion is a bit lack of scientific rigor.

Response:

Thanks for the reviewer's comments. Thank you for your seriously review. In view of your suggestions, we carefully check the full manuscript and calculation process again. The solution to the first suggestion is as follows:

In Table 6. The result of "Research Model" in the third line of the last manuscript was the fitting result of SEM model. Here we did not write clearly, and caused ambiguity, deeply ashamed. We have adjusted the form and text.

You mentioned lines 646-647 of the previous manuscript "As shown in Table 6, the same indices can be used to evaluate adequacy of both the CFA and SEM. All indices are in the appropriate range, indicating the model is acceptable. ” I'm sorry for the confusion caused by the expression and writing. We didn't mean that the result of CFA was used to represent the result of SEM.

It was probably a misunderstanding caused by the unclear expression and identification of Table 6.The indicators observed by CFA and SEM are the same indicators, but the measurement results of the indicators are different. The indicator results of SEM and CFA meet the fitting requirements.As for the second suggestion you mentioned, we have submitted it to the designated website and provided a link to download the source data as required by PLoS. Thank you for your advice.

And thank you for your 3rd suggestions. We have completed them one by one and added some references. Thank you for your careful guidance, and we will pay more attention in the future.

If there are still areas for improvement, we are willing to seriously revise them as soon as possible.

Once again, we sincerely thank you for your enthusiastic work, your careful evaluation and patient guidance. We learned a lot. It was a very pleasant experience.

We hope that the revision of the manuscript and our reply will be enough to make our manuscript suitable for publication in PLoS ONE.

We look forward to hearing from you at your earliest convenience.

Yours sincerely,

Lu Chen

---

## [Decision Letter · Decision Letter 3]

26 Jan 2022

A study on the influencing factors of the public's willingness to donate funds for critical illness crowdfunding projects on network platforms

PONE-D-21-03866R3

Dear Dr. Chen,

We’re pleased to inform you that your manuscript has been judged scientifically suitable for publication and will be formally accepted for publication once it meets all outstanding technical requirements.

Kind regards,

Barbara Guidi

Academic Editor

PLOS ONE

Additional Editor Comments (optional):

Reviewers' comments:

Reviewer's Responses to Questions

**Comments to the Author**

1. If the authors have adequately addressed your comments raised in a previous round of review and you feel that this manuscript is now acceptable for publication, you may indicate that here to bypass the “Comments to the Author” section, enter your conflict of interest statement in the “Confidential to Editor” section, and submit your "Accept" recommendation.

Reviewer #3: All comments have been addressed

2. Is the manuscript technically sound, and do the data support the conclusions?

Reviewer #3: Yes

3. Has the statistical analysis been performed appropriately and rigorously? 

Reviewer #3: Yes

4. Have the authors made all data underlying the findings in their manuscript fully available?

Reviewer #3: Yes

5. Is the manuscript presented in an intelligible fashion and written in standard English?

Reviewer #3: Yes

6. Review Comments to the Author

Reviewer #3: Many thanks for the authors to submit the revised version. I am fully satisfied with the responses and changes made by the authors.

7. PLOS authors have the option to publish the peer review history of their article (what does this mean?). If published, this will include your full peer review and any attached files.

Reviewer #3: No

---

## [Editor Report · Acceptance letter]

3 Mar 2022

PONE-D-21-03866R3 

A study on the influencing factors of the public's willingness to donate funds for critical
illness crowdfunding projects on network platforms 

Dear Dr. Chen:

I'm pleased to inform you that your manuscript has been deemed suitable for publication in PLOS ONE. Congratulations! Your manuscript is now with our production department. 

Kind regards, 

on behalf of

Dr. Barbara Guidi 

Academic Editor

PLOS ONE